

# Interwoven processes in fish development: microbial community succession and immune maturation

Lisa Zoé Auclert[1], Mousumi Sarker Chhanda[1,2] and Nicolas Derome[1]

[1] Département de Biologie, Institut de Biologie Intégrative et des Systèmes, Université Laval, Québec, Canada
[2] Department of Aquaculture, Faculty of Fisheries, Hajee Mohammad Danesh Science and Technology University, Basherhat, Bangladesh

## ABSTRACT

Fishes are hosts for many microorganisms that provide them with beneficial effects on growth, immune system development, nutrition and protection against pathogens. In order to avoid spreading of infectious diseases in aquaculture, prevention includes vaccinations and routine disinfection of eggs and equipment, while curative treatments consist in the administration of antibiotics. Vaccination processes can stress the fish and require substantial farmer's investment. Additionally, disinfection and antibiotics are not specific, and while they may be effective in the short term, they have major drawbacks in the long term. Indeed, they eliminate beneficial bacteria which are useful for the host and promote the raising of antibiotic resistance in beneficial, commensal but also in pathogenic bacterial strains. Numerous publications highlight the importance that plays the diversified microbial community colonizing fish (i.e., microbiota) in the development, health and ultimately survival of their host. This review targets the current knowledge on the bidirectional communication between the microbiota and the fish immune system during fish development. It explores the extent of this mutualistic relationship: on one hand, the effect that microbes exert on the immune system ontogeny of fishes, and on the other hand, the impact of critical steps in immune system development on the microbial recruitment and succession throughout their life. We will first describe the immune system and its ontogeny and gene expression steps in the immune system development of fishes. Secondly, the plurality of the microbiotas (depending on host organism, organ, and development stage) will be reviewed. Then, a description of the constant interactions between microbiota and immune system throughout the fish's life stages will be discussed. Healthy microbiotas allow immune system maturation and modulation of inflammation, both of which contribute to immune homeostasis. Thus, immune equilibrium is closely linked to microbiota stability and to the stages of microbial community succession during the host development. We will provide examples from several fish species and describe more extensively the mechanisms occurring in zebrafish model because immune system ontogeny is much more finely described for this species, thanks to the many existing zebrafish mutants which allow more precise investigations. We will conclude on how the conceptual framework associated to the research on the immune system will benefit from considering the relations between microbiota and immune system maturation. More precisely, the development of active tolerance of the microbiota from the earliest stages of life enables the sustainable establishment of a complex healthy microbial community in the adult host. Establishing a balanced host-microbiota

Corresponding authors
Lisa Zoé Auclert,
lisa.auclert.1@ulaval.ca
Nicolas Derome,
nicolas.derome@bio.ulaval.ca

interaction avoids triggering deleterious inflammation, and maintains immunological and microbiological homeostasis.

## INTRODUCTION

The intricate interplay between fish microbiota and their immune system is of paramount importance in understanding the development and maintenance of a healthy aquatic ecosystem (*Gomez, Sunyer & Salinas, 2013*). This review embarks on a unique journey, delving into the evolutionary relationship between microbiota and the fish immune system across various stages of development. Unlike previous studies, this review explores the dynamic interconnection between microbial communities and immune maturation during fish development.

The initial microorganisms (pioneering microbiota) play a pivotal role in shaping immune system development (*Zhang, Chi & Dalmo, 2019*; *Deng et al., 2021*). Thus, the health and survival during fish development involves two intricately connected processes: the assembly and progression of microbial communities, and the training of the immune system (*Stephens et al., 2015a*). Analyzing immune-microbiota interactions during fish development is essential for identifying patterns that support the transition to healthy adult stages, ultimately fostering the advancement of aquaculture practices (*Phelps et al., 2017*).

Among the efforts to control microbiota in aquaculture, since 1940, vaccination serves as a powerful tool in averting a diverse array of bacterial and viral diseases, targeting pathogenic microorganisms of the microbiota (*Ma et al., 2019*; *Yu et al., 2020*). Then, researchers have also successfully explored methods to not only eliminate pathogens, but also promote beneficial symbionts, with the first probiotic being used in 1986 (*Martínez Cruz et al., 2012*). Recently, the impact of various farming systems on the presence of beneficial microorganisms was also examined in an attempt to favour beneficial bacteria (*Vadstein et al., 2018*; *Deng et al., 2021*). The critical period of fish larval survival coincides with the recruitment of pioneering microbiota and host immune system development (*China & Holzman, 2014*; *Butt & Volkoff, 2019*; *Martínez et al., 2021*). Understanding the evolution of microbiota during early life stages is crucial, as it directly impacts growth, health, and overall survival (*Piazzon et al., 2019*; *Deng et al., 2021*). Pioneering microbiota significantly shape immune system development. Indeed, when fish are treated with compounds containing pathogen-associated molecular patterns, it activates non-specific innate immune mechanisms, resulting in later increased resistance to diseases, even before apparition of the adaptive response in larvae (*Zhang, Chi & Dalmo, 2019*; *Deng et al., 2021*; *Stephens et al., 2015a*). Analyzing immune-microbiota interactions during fish development is essential for identifying patterns supporting the transition to healthy adult stages, ultimately advancing aquaculture practices (*Phelps et al., 2017*).

This article's targeted audience includes researchers and professionals working in the fields of aquaculture, microbiology, ecology (microbial ecology), immunology and involved in fish farming who could gain insights into alternative disease prevention strategies to promote fish health, while minimizing the use of antibiotics.

## SURVEY METHODOLOGY

To write this review, we searched the research articles published between January 2001 and April 2023 indexed in PubMed and the Web of Science databases. The search terms typed were: Fish AND Microbiota for a first review information and then: immune system AND microbiota AND relations OR interactions AND fishes. The reference lists of these articles were also checked and included in the research. This review highlights recent progress in the knowledge of immune system and microbiota interaction in order to prompt the proposition of new potential therapeutic tools and sustainable practices in aquaculture.

### Fish immune system
#### Introduction on fish immune system
*Fundamental functions, and variations across species.* In this review, when we mention one organism's immune system (IS), we refer to all the structures involved in one or more of these three actions: recognition, degradation and elimination of pathogenic agents or diseases. It comprises the organs, cells and molecules involved in keeping a balance between immunogenicity and immune tolerance.

In fishes, an innate as well as an adaptive immune system (AIS) have been extensively described (*Secombes & Wang, 2012*). The innate immune system (IIS) is the result of a long-term memory on evolutionary scales and provides the generic response of an organism to diseases frequently encountered in the species evolutionary times (independently from the pathogens encountered by a specific individual) (*Rauta, Nayak & Das, 2012*). The IIS plays a major role in starting the first inflammatory response to an infection (*Sullivan et al., 2021*). The AIS is the system of responses adjusted after a first encounter with a pathogen by an individual organism (and whose activation is triggered by the innate immune system) (*Magnadóttir, 2006*). The AIS depends, for each organism, on its individual history of pathogens and diseases encountered and allows a quicker and more specific response in case of a second encounter occurring with the same pathogen (*Kurtz, 2005*; *Rauta, Nayak & Das, 2012*; *Kordon, Pinchuk & Karsi, 2021*).

Teleost fish are the oldest vertebrates with an adaptive immune system in addition to the innate immune system (*Whyte, 2007*). The adaptive component appeared in gnathostome vertebrates about 450 million years ago and enables the establishment of specific defense *via* the rearrangement of the V(D)J genes to allow the host to recognize a greater number of antigens (*Pancer & Cooper, 2006*; *Whyte, 2007*; *Weinstein et al., 2009*). Mutations in these genes can lead to dysfunctional DNA joining events or loss of the genes function, thus to immunodeficiencies in teleosts normally having their adaptive immune system (*Smith, Rise & Christian, 2019*; *Christie, Fijen & Rothenberg, 2022*). These rearrangements require the enzymes RAG-1/2 (Recombination Activating Gene 1), which became present after the integration of the RAG-1 and RAG-2 genes, characteristic of the adaptive immunity,

into the genome (*Bleyzac, Exbrayat & Fellah, 2005*; *Smith, Rise & Christian, 2019*). These genes are not present in every teleost species, some have lost the RAG genes and remained without this arm of adaptive immunity, like several angler fish (*Isakov, 2022*). In order to study the impact of the adaptive immune system on the microbiota in fish having the adaptive component, temporary inactivation of these genes is practiced, and fish mutants depleted for these genes were also developed to decipher interactions between adaptive immune system and microbiota not relying on the innate response (*Wienholds et al., 2002*; *García-Valtanen et al., 2017*; *Novoa et al., 2019*).

Fishes have different immune system structures and organs to interact with their microbiota, depending on their phylogeny (*Zapata et al., 2006*). Table 1 highlights differences among fishes with distinct immune organs and possession of RAG genes as indicators of adaptive immunity. As organisms in constant contact with water, fishes need to face a plethora of microorganisms and maintain balance between exclusion of harmful microbes while tolerating beneficial symbionts (*Yajima et al., 2022*). Over the course of evolution, their distinctions were indeed refined to align with the distinct spectrum of immune requirements for each species (reviewed in *Magadan, Sunyer & Boudinot, 2015*).

The lungfish living in chronically dry environments developed strategies to survive drought and protect against microorganisms through extreme increase in its mucus secretion and density, ultimately hardening and forming a protective cocoon (*Sturla et al., 2002*). This cocoon was demonstrated recently to contain different active cell types including immune cells and to display transcription of immune genes like antimicrobial peptide genes, cytokines, granulocyte activity markers occurring in the cocoon (*Heimroth et al., 2021*). It is reported that 76% of fish living in deep waters display bioluminescence, intrinsic or acquired *via* bioluminescent microorganisms (*Martini & Haddock, 2017*). Several of the marine fish depending on mutualistic symbionts for bioluminescence even developed specific organs: for examples Flashlight Fish (*Anomalops katoptron*) have suborbital light organs hosting luminous symbiotic bacteria producing light. Anglerfishes developed a specific organ (illicium) tolerating and selecting bioluminescent bacteria, and the function of the organs relying on the presence of the bacterial strains to emit light (*Freed et al., 2019*). The development and conservation of these organs indicates a co-evolution fish-microbiota and tolerance and selection mechanisms to target required symbionts, ultimately provides an evolutionary advantage to fish and these symbionts (*Tanet et al., 2020*).

It is also hypothesized that the jawed fish developed an adaptive immune system as they were probably exposed to more injuries and associated infections caused by their jaw structure and predator behavior favoring injuries compared to jawless fish (*Matsunaga & Rahman, 1998*).

The immune response mechanisms are constantly refined through time, taking into account the cost of the immune response and its benefits. For example, in three-spined sticklebacks, in response to a parasite infection, some evolved to resist and others to tolerate the harmful parasite as the resistance was associated to a reduced reproductive success (*Weber et al., 2022*). This article shows that immunity is not always better than

Auclert et al. (2024), *PeerJ*, DOI 10.7717/peerj.17051

**Table 1  Species wise differences in immune organs.** Sources: (a) *Savino & Santa-Rosa, 1982*; (b) *Araújo et al., 2019*; (c) *Kumar, Joy & Singh, 2016*; (d) *Nishimura, 1977*; (e) (*Mamun et al., 2022*); (f) *Goessling & Sadler, 2015*; (g) *Parra, Reyes-Lopez & Tort, 2015*; (h) *Churchill et al., 1979*; (i) *Johnson, Noe & Bauer, 1982*; (j) Swann et al., 2020 (k) *Baker et al., 2019*; (l) *Star et al., 2011*; (m) *Star et al., 2011*; (n) *Mao et al., 2015*; (o) *Gjessing et al., 2011*; (p) *Vildmyren et al., 2022*; (q) *Inami et al., 2009*; (r) *Yu et al., 2022*; (s) *Bracamonte et al., 2019*; (t) *Liu et al., 2022*; (u) *Brinkmann et al., 2004*; (ua) *Heimroth et al., 2021*; (ub) *Ojeda et al., 2006*; (uc) *Hiong et al., 2015*; (ud) *Mohammad et al., 2007*; (ue) *Tacchi et al., 2015*; (v) *Fänge & Pulsford, 1983*; *Schluter & Marchalonis, 2003*; (w) *Yu et al., 2020*; (x) *Lauriano et al., 2019*; (y) *Doane et al., 2020*; (z) *Mitchell & Criscitiello, 2020*. For Osteichthyes, indication of organ apparition chronology estimated for zebrafish is indicated by numbers (no. 1, no. 2) (*Stachura & Traver, 2011*; *Braunbeck et al., 2015*; *Bajoghli et al., 2019*).

| Fish | MALT's | Spiral valve | Thymus | Epigonal and Leydig organs | Spleen | Kidney | Liver | RAG 1/2 | Notes on immune-microbiotal specificities | Ref. |
|---|---|---|---|---|---|---|---|---|---|---|
| Osteichthyes | yes (no. 1) | no | yes (no. 3) | no | yes (no. 5) | yes (no. 3–4) | yes (no. 2) | yes, excepted few Anglerfish | Anglerfish: Special luring organ (illicium) dedicated to host bioluminescent microbiota recruited from the environment in a mutualistic interaction, the bacteria living in a protected environment, while the host takes advantage of the light for defense, attraction of preys and reproduction ; Some species (for example P. spiniceps and H. mollis) have lost key components of adaptive immunity, with alterations in rag-1 and rag-2 genes (this is often associated to sexual parasitism like permanent parabiosis). Gadidae: do not have Major Histocompatibility Complex-II nor CD4 proteins, thus has a week response to vaccines targeting the adaptive immune system via the MHC-II. Lungfish: cocoon produced to survive dry seasons on land, contains live immune cells: granulocytes, provides protection from infections. | (a) (b) (c) (d) (e)(f) (g)(h) (i)(j) (k) (l)(m) (n)(o) (p)(q) (r)(s) (t) (u)(ua) (ub) (uc) (ud) (ue) |
| Chondrichthyes | yes | yes | yes | yes | yes | yes | yes | yes | Spiral Valve containing contains lymphoid aggregates; antibodies without L chains in nurse sharks. Elasmobranchii exhibit phylosymbiosis. | (v)(w) (x)(y) |
| Agnatha | yes | no | no | no | yes | yes | yes | no | No thymus but a thymoïd region in lampreys. | (z) |

susceptibility to infection if the cost of enduring the infection is less significant than that of immunological resistance.

In this review, we will mainly focus on teleost fish if not otherwise mentioned, and especially on few species of particular interest and for which more studies were undertaken to both characterize their immune system and microbiota: salmonids, carps and zebrafish (an experimental model very suitable for understanding the ontogeny of the immune system because of many mutants available and quick growth rate) (*Zapata et al., 2006*).

*Components of the immune system in adult teleost fishes.* We will briefly present the main organs of the immune system, associated cells and function in the Table 2. The main lymphoid organs of teleosts are the thymus, the kidney and the spleen and they also present crucial mucosal associated lymphoid tissues (MALTs), other reviews presented in more exhaustive details teleosts immune system (*Holland & Lambris, 2002*; *Lam et al., 2004*; *Levraud & Boudinot, 2009*; *Rauta, Nayak & Das, 2012*). Then, we will focus on the ontogeny of the immune system. This will enable to highlight in part 2, Fish microbiota, how the recruitment of particular microbial symbionts is enabled at specific steps and stimulates the development of the immune system, by excluding specific strains while favoring others.

**Cellular innate immunity**

Macrophages and dendritic cells have the capability to phagocyte and then activate naïve T lymphocytes thus linking the IIS to the AIS. Macrophages and dendritic cells have been described in zebrafish, Atlantic salmon and carp species (*Pettersen et al., 2008*; *Lugo-Villarino et al., 2010*, p.; *Renshaw & Trede, 2012*; *Imanse et al., 2022*). Other cells display natural killer functions (NK), and neutrophils, mast cells, basophils, eosinophils are also active in the zebrafish, salmon and carp species (*Shah et al., 2012*; *Haugland, Jordal & Wergeland, 2012*; *Shim et al., 2019*; *Megarani et al., 2020*; *Castranova et al., 2022*).

**Cellular adaptive immunity**

In zebrafish AIS, the main T lymphocytes characterized in mammals are present: lymphocytes killers CD8+ (LTk), helpers CD4+ (LTh) and regulatory CD4+CD25+ (Ltreg) (*Renshaw & Trede, 2012*). LTk, when activated, eliminate infected and abnormal cells (such as cancerous cells) by recognizing aberrant CMH-I presentation with their CD8+ serving as the receptor for CMH-I recognition. LTh bear the CMH-II recognizing receptor, the CD4+, these lymphocytes help the activation of other cells by secreting cytokines to dampen (Th17), or increase inflammation (Th1), and contribute to the activation of LTk and memory B cells. Then, the LTreg are involved in tolerance (are also called suppressive lymphocytes as they tend to prevent excessive inflammation) (*Nakanishi, Shibasaki & Matsuura, 2015*; *Scapigliati, Fausto & Picchietti, 2018*; *Ashfaq et al., 2019*, p. 4). Finally, B cells are producers of antibodies (*Ye et al., 2013*).

**Molecular immunity**

These cells interact between them and with pathogens *via* a complex network of pro- or anti-inflammatory molecules as well as antimicrobial molecules. These include cytokines, antimicrobial peptides (AMPs), proteases, lysozymes, lectins, molecules of the complement system and the pattern recognition receptors (PRRs) recognizing damages

**Table 2** Overview of cell types present in the main teleosts immune organs, and associated functions. References : (a) *Ángeles Esteban, 2012*; *Dash et al., 2018*; *Almeida et al., 2019*; (b) *Salinas, 2015*; *Kelly & Salinas, 2017*; *Dalum et al., 2021*; (c) *Fuglem et al., 2010*; (d) *Salinas, 2015*; *Gaudino & Kumar, 2019*; *Rességuier et al., 2017* (e) *Mokhtar & Abdelhafez, 2021*; (f) *Holden, Layfield & Matthews, 2013*; (g) *Harvie & Huttenlocher, 2015*; (h) *Whyte, 2007*; (i) *Rauta, Nayak & Das, 2012*; (j) *Geven & Klaren, 2017*; (k) *Muire, 2017*; (l) *Dowling & Hodgkin, 2009*; (m) *Shwartz, Goessling & Yin, 2019*.

| Organ, Localisation | Immune Cell types | Function | Ref. |
|---|---|---|---|
| Mucosa associated lymphoid tissues (MALTs) of skin, gilll, gut, nasopharynx | Goblet cells, sacciform cells and club cells | Continuous mucus secretion that hinders the attachment of pathogens and concentrates antibacterial molecules (lysozyme, antibodies, complement proteins, lectins, C-reactive protein, lytic enzymes and other antibacterial peptides), thus providing a first physico-chemical barrier against pathogens. Mucus entraps some commensal bacteria and bacteriophages, adding protection through the phenomenon of resistance to colonisation and targeted antibacterial phage effect. | (a) |
| | T lymphocytes and B lymphocytes | Accumulation of T lymphocytes at interbranchial lymphoid tissue (ILT), CD8+ T lymphocytes exhibit a cytotoxic phenotype, and tissue resident memory T cells enable a locally adapted immunity in the MALTs. Diffuse presence of B cells that express the antibody IgT/Z. | (b) |
| | Antigen-presenting cells (APCs) | APCs include macrophages, dendritic cells (these cells are more present in fish surfaces exposed to microorganisms), B cells that have antigen-sampling abilities and interact with T cells, but nee to be studied more in detail in teleost fishes. | (c)(d) |
| | Leucocytes: Granulocytes, eosinophils, neutrophils | Granulocytes produce immunoreactive inflammatory iNOS2 during inflammation or stress, which induces a cytotoxic environment. Neutrophils are quickly recruited at MALTS (from kidney) during infection and degranulate (granules containing antimicrobials), phagocyte microorganisms, produce of ROS, and release Neutrophil extracellular traps (NETs) that are networks of external DNA fibers from the neutrophils that entraps pathogens). | (e) |
| | M-like cells | Antigen-sampling abilities, characterized in rainbow trout. | (c) |
| Kidney | Hematopoietic cells | Fish do not have a bone marrow; the hematopoiesis occurs in the head kidney and in the posterior kidney. | (f) |
| | Lymphoid cells | T Lymphocytes in interaction with B lymphocytes. Proliferation, differentiation of B lymphocytes in the head kidney and production of antibodies (The primary site for antibody production in teleosts is the head kidney). | (e) (f) |
| | Macrophages | Macrophages are produced in kidney, and do antigen phagocytosis and degradation, tissue reparation, can synthesize cytokines and growth factors, can gather forming melano-macrophage centers (MMCs) in headkidney, MMC produces IgM and contribute to immune memory (forming memory cells) by keeping the antigens and through antigenic presentation. | (e)(h) |
| | Neutrophils | Neutrophils are produced in kidney and then disseminated via blood circulation, they have hydrolytic enzymes, lysosomes and phagosomes and cytotoxyc granules in their cytoplasm, are quickly recruited at infection sites to degranulate (granules containing antimicrobials), phagocyte microorganisms, produce of ROS, and release Neutrophil extracellular traps (NETs) that are networks of external DNA fibers from the neutrophils that entraps pathogens) . | (e)(g) |

**Table 2** (*continued*)

| Organ, Localisation | Immune Cell types | Function | Ref. |
|---|---|---|---|
| | Endocrine cells (steroidogenic cells) | Cortisol is secreted to adapt the systemic inflammation adequately for the fish survival, when the hypothalamic-pituitary-interrenal (HPI) axis is activated. Microbiota composition can affect fish stress and cortisol release, and cortisol influences fish microbiota composition. | (f) (i) (j) |
| | Natural killer cells | Are produced in kidney, spleen, and liver, enable elimination of non-self cells and cells intracellularly infected by a pathogen. | (k) |
| Thymus | Macrophages | Phagocytosis and breakdown of antigens, tissue reparation, can synthesize cytokines and growth factors, can gather forming MMCs. They enable antigen presentation, this organ being crucial for early setting up of the adaptive immune system. | (e) (h) |
| | B and T Lymphocytes, thymocytes ,Natural Killers | Thymus is site of antigen presentation in fish and creates mature T-cellular pool adapted to then settle in peripheric MALTs, being adapted to fight the environmental pathogens met in the individual's environment. | (l) |
| Spleen | Macrophages | Antigen phagocytosis and degradation, tissue reparation, can synthesize cytokines and growth factors, can gather forming MMCs.In the spleen, antigen presentation occurs, linking innate to adaptive immune response. | (e) |
| | Leucocytes | Produce immunoreactive iNOS2 in context of inflammation in MMCs of the spleen | (e) |
| | Lymphocytes | Spleen contains a lot of mature B lymphocytes (plasmocytes) producing antibodies. T Lymphocytes also interact with antigen presenting B cells making the spleen also an organ linking innate to adaptive immunity. | (i) |
| | Natural killer cells | Are produced in kidney, spleen and liver, enable elimination of non-self cells and cells intracellularly infected by a pathogen. | (k) |
| Liver | Macrophages | Liver filters the blood stream coming from the fish gut (containing food, microorganisms, and their metabolites) to remove potentially harmful components. The liver hosts hepatic resident macrophages which are crucial for maintaining immunological functions. During homeostasis, the resident macrophages of the liver eliminate by phagocytosis the pathogens from the blood. | (m) |
| | Natural killer cells | Are produced in kidney, spleen, and liver, enable elimination of non-self cells and cells intracellularly infected by a pathogen. | (k) |

*via* damage-associated molecular patterns (DAMPs) or pathogen molecular motifs *via* pathogen-associated molecular patterns (PAMPS): toll-like receptors (TLRs), NOD-like receptors (NLRs, extensively described in *Palti, 2011*; *Zhang et al., 2018*), immunoglobulins, Tcell receptors (TCRs), Bcell receptors (BCRs), maintaining a defense against pathogens and diseases in healthy animals (*Holland & Lambris, 2002*; *Whyte, 2007*; *Beutler, 2009*).

The innate receptors (NLRs, TLRs) recognise pathogen or danger associated molecular patterns (PAMPs and DAMPs) not present in healthy tissues. These molecular patterns were encountered by the species for a long evolutionary time in association with damage to host tissues, thus leading to strong selection of innate receptors to fight against such prevalent pathogens. These receptors detect frequently encountered pathogen components like LPS, bacterial or viral RNA (*Magnadóttir, 2006*).
The lysozyme is an enzyme often studied from the fish innate immunity, targeting bacterial peptidoglycans and which activity allows the bacterial cell lysis (*Smith, Rise & Christian, 2019*). In teleost fishes, there are only three types of antibodies, the two types of mucosa associated antibodies: the IgM tetramer and the fish-specific IgT/IgZ monomers (*Bilal, Etayo & Hordvik, 2021*). Finally, IgD, whose role is not yet fully understood, is abundant in gills but not in the intestine nor in peripheral blood (*Ye et al., 2013*). In healthy animals, inflammation is often necessary to fight pathogens efficiently, but must be controlled to avoid excessive or chronic effect. Indeed, a healthy state requires immune homeostasis: inflammation must reach a minimum threshold to fight efficiently against invading pathogens, abnormal cells and tissues, but stay below a maximum threshold to avoid inducing excessive inflammation that could damage healthy tissues. When immune equilibrium is disturbed, the fish's immune response may become inactive or overactive, any of those situations may lead to sickness, either by absence of immune response or by critical inflammation.

The above-described immune system features are in place at the adult stage, resulting from a temporal sequence during which the immune system develops and matures progressively as does the whole individual's body. Therefore, immune system in recently hatched fish larvae is immature. Fishes are ectothermic animals and develop quicker at higher temperatures (in the range of tolerated temperatures by each species), as does their immune system (*Stolen et al., 1984*; *van Denderen et al., 2020*; *Feidantsis et al., 2021*). Therefore, temperature must be tightly controlled when studying the temporal development of the fish's immune system (*Cooper, 1985*; *Zapata et al., 2006*). For zebrafish, both temperature and the inter-individual variability were observed to control immune system ontogeny: functional adaptive immune system occurred between 35 to 75 days post fecundation (dpf) (*Stephens et al., 2015a*).

We will now describe some chronological key steps of fish immune system development that are important for the host-microbiota interaction.

### Ontogeny of the immune system

*A first autologous molecular immunity.* Initially, and for a relatively short period of time after fertilization (*i.e.*, blastula stage), only inherited factors are involved in development and immune protection, as the new genome is not yet active (*Lee et al., 2013*). During the maternal-to-zygotic transition (MZT), the zygote genome gradually leaves its quiescent state to allow the individual's own embryonic genes to start expressing themselves and thus producing their own (autologous) immune defenses (*Lee, Bonneau & Giraldez, 2014*). The time to get the genome activated varies greatly between species. For the zebrafish, the increase in genomic transcripts corresponding to the nuclear genome activation occurs between 10 and 12 cellular cycles at approximately 3 h post fertilization, (*Aanes et al., 2011*; *Lee et al., 2013*; *Lee, Bonneau & Giraldez, 2014*; *Kotani, Maehata & Takei, 2017*). This MZT is believed to start at same cellular cycle (10th) and thus at different times post fertilization in other oviparous fishes like rainbow trout for example (*Ocalewicz et al., 2019*). Indeed, rainbow trout reaches a 64-cell stage only after 21 h or during mid-blastula (*Ramachandra et al., 2008*), so it takes more than a day to reach the MZT and start expressing its own

genes, including those involved in the immune response (*Knight, 1963*; *Ramachandra et al., 2008*; *Ocalewicz et al., 2019*).

Following MZT, genes involved in the innate immune response are first expressed. These early autologous defenses include the pattern recognition receptors, molecules like the lysozyme degrading the bacterial peptidoglycan, lectins. Few studies have focused on the non-cellular components of innate immunity, which are known to appear before hatching (*Magnadóttir, 2006*; *Li et al., 2011*) although the precise timing of their appearance remains to be discovered.

In rainbow trout, before hatching, the intelectin expression starts to increase at 5 days post fertilization and then significantly from 13 dpf to 26 dpf. The autologous expression of the secreted TLR5 targeting bacterial flagellin is also significantly increased at 5 dpf, while the mTLR5 levels stayed stable. Lysozyme activity increases during the first dpf, thus also demonstrating an autologous expression from the rainbow trout embryos. This highlights that newly fertilized eggs express (non-cellular) innate immune defenses quickly after fertilization and MZT, during the first cell divisions (*Li et al., 2011*).

In zebrafish, studies have shown that the expression of intestinal alkaline phosphatase (IAP) is higher before 5 days post fertilization. This allows the larvae to tolerate a high concentration of LPS. However, by 6 days post fertilization, the same concentration of LPS becomes lethal for the fish (*Bates et al., 2007*). This could denote a first-life stage in which the recruitment of microorganisms, including the bacteria presenting LPS, is more permissive and leads to less inflammatory responses. In addition, some studies observed TLRs and MyD88 expression during early embryo stages, then increasing along the developmental stages (*Jault, Pichon & Chluba, 2004*; *Meijer et al., 2004*). Innate immune receptors like PAMPs and DAMPs, have an increasing expression throughout the life (*Meijer et al., 2004*).

*Immune organs ontogeny and cellular innate immunity.* For zebrafish, organs ontogeny happens in chronological order starting with the thymus, followed closely by the kidney, and ultimately the spleen, while the ontogeny and apparition times for MALTs still need to be investigated (*Stachura & Traver, 2011*; *Bajoghli et al., 2011*; *Ángeles Esteban, 2012*; *Muire, 2017*). In zebrafish, the kidney is the first organ where hematopoietic precursor cells can be found, but the thymus is the first organ containing immune lymphoid cells, and the spleen is the last organ bearing lymphoid cells (*Zapata et al., 2006*). This chronology seems conserved for the freshwater teleosts (*Zapata et al., 2006*). However, for marine species like cod, for example, the order of ontogenesis is slightly different, with kidney then spleen developing before the thymus (*Zapata et al., 2006*).

Zebrafish head kidney is present at 72 h post fertilization (hpf), and head and posterior kidney formation is complete at 4dpf, with efficient hematopoiesis at 6dpf (*McKee & Wingert, 2015*; *Poureetezadi & Wingert, 2016*; *Lucon-Xiccato et al., 2023*; *The Zebrafish Information Network, 2024*) Initially, hematopoiesis occurs in an intraembryonic locus, the intermediate cell mass (ICM), then after kidney formation, it becomes the main site where hematopoiesis takes place (*Zapata et al., 2006*; *Astin et al., 2017*). However, in zebrafishes,

lymphocytes were identified only after 2–3 weeks post fertilization, which coincides with the reported detection of Rag transcripts (*Hansen & Zapata, 1998*; *Lam et al., 2004*).

The zebrafish thymic primordium is formed at 60 hpf and 5 h later (65hpf) is containing immature lymphoblasts (*Willett et al., 1999*). Thymus and kidney are in development at 72 hpf and already contain markers for lymphoid progenitors (*Willett et al., 1999*; *Trede et al., 2004*; *Miao et al., 2021*). The thymus will differentiate and contain only after 21 dpf the first CD8+ and CD4+ lymphocytes, marking the onset of the adaptive immunity (*Willett et al., 1999*).

The spleen is the main organ where adaptive immune responses are generated, after 2–3 weeks post fertilization (*Hansen & Zapata, 1998*; *Lam et al., 2004*). But the starting time of splenogenesis still needs to be investigated (*Bjørgen & Koppang, 2021*; *The Zebrafish Information Network, 2024*).

Early macrophages were observed in zebrafish as soon as 15 hpf, described as efficiently functioning at 30 hpf and contributing to the constitution of non-inflammatory macrophages residing in the brain and epidermis, patrolling and constantly circulating between epithelial cells. Granulocytes (neutrophils and eosinophils) circulate by 48 hpf. In zebrafish, the myeloblasts have been identified entering the circulation by 34 hpf. (*Herbomel, Thisse & Thisse, 2001*; *Lieschke et al., 2001*; *Astin et al., 2017*).

To compare between species, it can be more relevant to compare fish's developmental stages rather than days of life. Indeed, depending on the species under consideration, the growth rate and development of fish and of their immune response may vary in terms of the time required to be efficient. However, many processes follow a similar chronological sequence, and the onset of cellular innate immunity generally precedes the immune organs ontogeny, which precedes adaptative cellular immunity and the apparition of the humoral immune response in teleosts (*Zapata et al., 2006*). Development times of immune lymphoid organs ontogeny are provided in Table 3 for different teleosts: zebrafish rainbow trout, Cod and Carp.

### Adaptative immune response

The expression of the Rag-1 gene (Recombination activating gene 1, involved in V(D)J recombination) is crucial to activate the adaptive immune system (and also the expression of activation-induced cytidine deaminase (AID) that in fish catalyzes somatic hypermutation but not class switching as the class switching does not happen in fish) (*Stavnezer & Amemiya, 2004*). It was described that the expression of the Rag-1 gene can be detected as soon as 8 dpf in the pronephros of zebrafish, interpreted as indicating its expression in immature lymphocytes from pronephros in zebrafish (*Trede et al., 2004*). Indeed the observation of effective lymphocytes in zebrafish is possible only after 2 or 3 weeks post fertilization (*Trede et al., 2004*). To study the effect of a lack of adaptive immune system, Rag1-/- zebrafish mutants are available, that are viable but more fragile to infections. However, Rag1-/- zebrafish's innate system was demonstrated to compensate for the lack of adaptive immunity by enhancing expression of innate immunity-related genes, thus the mutation Rag1-/- impacts also indirectly the regulation of the innate immune system in these mutants (*García-Valtanen et al., 2017*). This underlines the interest of being able

Auclert et al. (2024), PeerJ, DOI 10.7717/peerj.17051

**Table 3  Species temporal variations in immune organ ontogeny among teleosts.** References: (a) *The Zebrafish Information Network, 2024*; (b) *Sahoo et al., 2021*; (c) *Schrøder, Villena & Jørgensen, 1998*; (d) *Sahoo et al., 2021*; (e) (f) *Schrøder, Villena & Jørgensen, 1998*; (g) *Huttenhuis et al., 2006*; (h) *van Loon, van Oosterom & van Muiswinkel, 1981*.

| Spp. | Thymus | Kidney | | Spleen | Ref. |
|---|---|---|---|---|---|
| Zebrafish | Tyymic primordium 60 hpf-65 hpf<br><br>2,5-6 dpf: organ fully developped | Headkidney 3–6 dpf<br><br>Posterior kidney 1–4 dpf | 3–6 dpf efficient hematopoiesis | Splenic primordium apparition: 5 dpf. Full development not documented. But apparent at 90 dpf | (a) (b) |
| | After 21 dpf: contains mature lymphocytes | | after 2–3 wpf: presence of lymphocytes | | |
| Rainbow trout | Thymus appears before hatching, actively lymphoid at hatching | Headkidney appears before hatching, actively lymphoid at hatching | | Splenic primordium at 3 dph | (c) (d) |
| Cod | Visible in 9mm larvae: 28 days after hatching | Headkidney present at hatching | | Spleen present at hatching | (e) |
| Carp | Appearance of the thymus at 3 dpf for common carp | in Carp, plasmocytes detected after 1 month post fertilization in pronephros and mesonephros | | visible from 10 dpf | (f) (g) (h) |

to generate temporary knockouts of Rag-1 in order to get rid of this compensation effect (*Trede et al., 2004*; *Novoa et al., 2019*, p. 1).

## Fish microbiota
### Fish microbiota and its functions
*Introduction to the microbiota and its functions.* The term "fish microbiota" refers to the complete set of microbes living in association with the fish. Microorganisms are found on fish body surfaces (such as skin, gills, mouth, genital papilla, *etc.*) and also inside of internal organs like the gut, liver and kidney (*Meron et al., 2020*; *Huang et al., 2020*). It includes numerous species of bacteria, archaea, unicellular eukaryotes and viruses. These microbial communities differ according to the site where they are hosted (organ, body region) (*Merrifield & Rodiles , 2015*).

The microbiota has various roles, such as facilitating nutrient absorption and digestion (*Semova et al., 2012*), contributing to the gut development (*Luan et al., 2023*), training the immune system (*Zhang, Chi & Dalmo, 2019*) and safeguarding against pathogens (*Verschuere et al., 2000*). An imbalance in microbiota composition, known as dysbiosis, can lead to the onset of various diseases (*Ley, Peterson & Gordon, 2006*; *Rawls et al., 2006*; *Hooper, Littman & Macpherson, 2012*; *Ye & Rawls, 2021*).

The bacterial microbiota is the most widely studied component, and is therefore the one the review will focus on. Characterization of the other components is still in its infancy (*e.g.*, archaea are only occasionally mentioned but not studied in detail) (*Rawls, Samuel & Gordon, 2004*). Still, an article comparing the intestinal mycobiota of wild with that of artificially raised zebrafish (*Siriyappagouder et al., 2018*), revealed that the dominant phylum was Ascomycota, and that resident fungal genera were able to produce crucial enzymes for fish nutrition (*Banerjee & Ghosh, 2014*; *Siriyappagouder et al., 2018*). Bacterial diversity decreases throughout the life stages in artificially raised zebrafish and in wild salmonids, and is low in wild adult salmonids (*Llewellyn et al., 2016*; *López Nadal et al., 2020*; *Rasmussen et al., 2023*). It could be hypothesized that the decrease in bacterial diversity corresponds to an increasing predominance of other non-bacterial microorganisms, or of one specific bacteria. For example, it was demonstrated that one single bacteria species, mycoplasma, can dominate the microbiota of adult Atlantic salmon (*Heys et al., 2020*); *Rasmussen et al., 2021*; *Rasmussen et al., 2023*; *Cheaib et al., 2021b*). The significance of other microorganisms, possibly from different kingdoms such as Archaea, fungi, or viruses, may be underestimated due to limited investigation. Therefore, more research including other microbes is required to gain a more comprehensive view of the microbiota. Recent findings described diverse fungal phylotypes in Atlantic salmon, at the first stages of development, providing a basis to understand the mycobiome associated with healthy management of fungal communities in Atlantic salmon farming (*Lokesh, Siriyappagouder & Fernandes, 2023*).

The composition and abundance of the microbial community are both influenced by many factors, abiotic or biotic. Abiotic factors include the temperature, presence of antibiotics or residual antibiotics in the water, fish diet, and salinity (*Boutin et al., 2014*; *Li et al., 2022*; *Llewellyn et al., 2016*; *Standish, Brenden & Faisal, 2016*; *Sylvain & Derome,*

*2017*; *Sylvain et al., 2019*; *Vadstein et al., 2018*). Salinity is a crucial factor as it was described as the main one driving the initial microbial community composition in environmental water, being a good explicative variable of variability in diversity observations (for archaeas and bacteria) in freshwater compared to saltwater environments (*Wang et al., 2012b*). In teleosts fish, it was demonstrated that the salinity is of paramount importance in the microbiota composition with, for example, studies on Asian sea bass (*Morshed et al., 2023*), Atlantic salmon (with increase in bacterial diversity when transferred to saltwater) (*Perry et al., 2020*), and Mediterranean sparids (*Scheifler et al., 2022*).

Biotic factors comprise the environmental microorganisms (*e.g.*, bacterioplankton) in contact with the fish, the vaccines administered, the host genotype and the fish social behaviour as fish in groups have been described to display a herd immunity, including microbiota exchanges between fish and their symbiotic hosts like anemones, as seen in clownfish (*Boutin et al., 2014*; *Émie et al., 2021*; *Standish, Brenden & Faisal, 2016*; *Sylvain et al., 2019*).

The recruitment and modulation of the microbiota is subject to several stochastic (genetic drift and dispersion) as well as determinist forces (competitive interactions, host filtering, which is adaptation to the microenvironments provided by the host). The contribution of these forces varies throughout the fish developmental steps, as the microbiota establishes itself and modifies locally its environment, influencing the development of the host and the differentiation of its immune system (*Derome & Filteau, 2020*). The microbiota changes considerably in the early stages of development, and at transition stages (at key steps of immune development and at the moments of drastic environmental changes for example for anadromous species). These early microbial communities have a determinant impact on fish immune fitness at the time of establishment (larvae), but also on later stages (influencing the immune responses through the imprinted modifications it produced during the formation of the immune system) (*Rudi et al., 2018*). Specific stages of the host immune system development allow the recruitment of specific stage-associated microbiotas based on the interaction between microbes and fish host immune cells and molecules (*Stephens et al., 2015a*).

*Approaches for exploring fish microbiota.* To study the microbiota, the most common approach is to use amplicon sequencing of taxonomic markers (*i.e.,* segments that are ubiquitously found in a group of microorganisms; for example, 16S ribosomal RNA for bacteria). It is then possible to study variations in fish gene expressions across different microbiota patterns corresponding to the conditions tested, using RNA-Seq (*Méndez-Pérez et al., 2020*; *Ofek et al., 2022*; *Fei et al., 2022*). Also, shotgun sequencing of microbial DNA can be performed to characterize the metagenome, which consists in the pool of microbial genes. It enables both taxonomic and functional annotations, although not allowing to reach the same depth as 16S sequencing and being more time and resources consuming. Additionally, the challenge of host genome amplification can result in a reduction in the number of microbiota reads. Recent studies also explored fish-microbiota gene interactions using dual RNA-Seq (*Westermann, Gorski & Vogel, 2012*; *Le Luyer et al., 2021*; *Sylvain et al., 2023*).

These approaches aim to give exhaustive description of the various microorganisms present, and enable to look for molecular signatures in communities associated with any given status of fish health (*Jovel et al., 2016*). One significant challenge after gathering the information, is to understand the evolutionary relationships between hosts and their microbiota. However, it can be difficult to decipher precise interactions that are crucial to their survival between the host and specific microbes. In this regard, the complexity of microbial community and contamination by host fish DNA can make it difficult to isolate and do precise identification and analysis of microbial genetic material challenging (*Rasmussen et al., 2023*). In addition, dual RNA-Seq comes with the challenging depletion of rRNA in both host and microbiota transcriptomes, which necessitates significant sequencing depth to get consistent reads counts for microbial transcripts.

The study of axenic (also called germ-free) fish models enables to decipher and demonstrate the crucial roles of the microbiota. Many axenic fish models are described in the literature, including in particular salmon, tilapia, sea bass, turbot and trout (*Gómez de la Torre Canny et al., 2023*; *Pérez-Pascual et al., 2021*).

Axenic models demonstrated that the microbiota improves the nutrition, immune response (*Galindo-Villegas et al., 2012*), neutrophil recruitment and gut epithelial cell renewal, and overall functioning of the fish gut (*Parker et al., 2018*). Microbiota encoded enzymes can help break down food that the host's own could not digest, and thus produces nutrients (vitamins). Microbiota encoded antimicrobial compounds (*e.g.*, bacteriocins) and/or various mechanisms of colonization resistance (established microbiota community protects itself and its host against intrusion of new microorganisms, including pathogens) can help protect the host against harmful germs and infection (*Derome & Filteau, 2020*; *Pérez-Pascual et al., 2021*; *Stressmann et al., 2021*). In axenic fishes, the gut epithelium does not differentiate properly (there is an absence of activity of intestinal alkaline phosphatase, immature patterns of glycan expression, and a lack of goblet and enteroendocrine cells) (*Bates et al., 2006*).

While cesarean section delivery is a common method for establishing axenic mammals, it can be used for viviparous fish but is not the only method to obtain axenic fish for oviparous species, due to their distinct reproductive biology. In these instances, the eggs can undergo thorough disinfection, rendering them axenic before hatching (*Pham et al., 2008*; *Smith, McCoy & Macpherson, 2007*).

These models can then be colonized with precisely engineered synthetic microbial communities (SynComs) or by one micro-organism to determine its effect on the host (*Stressmann et al., 2021*). Fish immunity-related genes can also be altered to observe their impact on the microbiota in gnotobiotic animals (*Stagaman et al., 2017*; *Stagaman, Sharpton & Guillemin, 2020*; *Xia et al., 2022*). Limitations are associated to the use of axenic model. Axenic rearing techniques developed for zebrafish to date have a quite short time limitation due to the difficulty to provide an adequate nutritious and axenic diet (the fish starts feeding at 5dpf, and can survive up to 15dpf unfed or if insufficiently fed). Indeed, the sterilization leads to an inaccessibility of the vitamins normally produced by the microbiota, which limits the survival time of axenic animals if no vitamin complementation is provided (*Fiebiger, Bereswill & Heimesaat, 2016*). Experiments showed no significant

difference in body length nor weight between axenic and non-axenic *Onchorynchus mykiss* nor anatomical differences in intestine, gills, brain, spleen and head kidney (*Jiménez-Reyes, Yany & Romero, 2017*; *Pérez-Pascual et al., 2021*), but a decrease in goblet cells was observed in axenic fish, as also reported in zebrafish (*Pérez-Pascual et al., 2021*; *Rawls, Samuel & Gordon, 2004*). In axenic Atlantic salmon fry compared to conventionally raised fry, a reduction in the mucus barrier was observed, and fewer oil globules were observed under axenic condition, indicating an impact on lipid metabolism (*Gómez de la Torre Canny et al., 2023*). However, the body size does not vary significantly between axenic and conventional fry (*Gómez de la Torre Canny et al., 2023*). In threespine stickleback, neutrophils were found to be less abundant (identified by their myeloid peroxidase activity) in axenic than in conventional fish having a microbiota, for 12 out of 17 stickleback families (*Milligan-Myhre et al., 2016*). Some microorganisms that are part of the fish microbiota were demonstrated to be able to secrete molecules, for example butyrate having anti-inflammatory effect (reducing neutrophil and macrophage recruitment and proportion of inflammatory macrophages expressing TNF $\alpha$), on the fish, thus promoting tolerance (of the micro-organisms producing the molecule and nearby organisms) (*Cholan et al., 2020*).

### Fish microbiota ontogeny

*First recruitment of early microbiota after hatching.* For the oviparous fishes, hatching larvae are suddenly exposed to all the environmental microorganisms from the surrounding water and colonized by the pioneer microbial community (*Sylvain & Derome, 2017*; *Murdoch & Rawls, 2019*). At this moment the immune system is still immature and presents only innate immune components that will constitute, joined with the fish surface physiological characteristics, the first filter to microbiota recruitment, and allowing the early recruitment of pioneer species (*Hitzfeld, 2005*). The first microbial community associated to the fish is sourced from the environmental microbial community, but only a fraction of species will be able to thrive in host body surfaces, depending on the host's physiology (local physical and chemical properties) and innate immune response (*Yan et al., 2016*). Interestingly, when testing offspring of parents infected with salmon gill poxvirus, it was observed that vertical transmission of this virus is not significant, while horizontal transmission was highly efficient. Therefore, egg membrane could provide a physical protection against the transmission of some pathogens from parents to offspring (*Norwegian Veterinary Medicine, 2024*; *Tørud et al., 2020*; *Gulla et al., 2020*).

In microbial ecology, this phenomenon is termed as selective colonization ability (*Zhou et al., 2019*), which belongs to a broader concept, the environmental filtering (*Kraft et al., 2015*). Indeed it has been described in zebrafish larvae guts at early stages (4dpf) that $\gamma$-proteobacteria are highly abundant, and $\beta$-proteobacteria are the second most abundant species, while the most abundant species found in the surrounding water samples were $\beta$-proteobacteria, followed by $\gamma$-proteobacteria (*Stephens et al., 2015a*; *López Nadal et al., 2020*).

For viviparous-like species as pipefish (*Syngnathus typhle*), and in viviparous species like kelp perch (*Brachyistius frenatus*), microbiota is transmitted prior to birth by the parents and microbial species are then recruited after birth from the environment (*Beemelmanns et*

*al., 2019*; *Boilard, 2021*). Additionally, continuous re-administration of the early microbiota can be done for some species by the parents interacting with the offspring at early stages by the parenting behaviour, this was described for cichlid fishes and transmission explanation were proposed to be induced by mouthbrooding behaviours or fry feeding by parental cutaneous secretions (*Sylvain & Derome, 2017*; *Keller et al., 2018*; *Spagopoulou & Blom, 2018*).

*Ecological succession and microbiota reorganization.* After the first microbial recruitment, as soon as the pioneer species are recruited, they will themselves modify their local environment chemically and physically and affect the immune development of the fish. The establishment of new species in a community depends on the order and timing of their arrival, this phenomenon is called the priority effect (*Derome & Filteau, 2020*; *Debray et al., 2022*). The bacterial biofilm will modulate the physico-chemical properties of the colonised body surfaces, quickly affecting the microenvironment (gills, intestine, skin, *etc.*) and facilitates the recruitment or exclusion of subsequent microorganisms (*Wahl et al., 2012*). The succession of microorganisms and the importance of the successive steps, each having a determining effect on the next step, can be referred as the microbial ecological succession, phenomenon diagrammed in the Fig. 1 (*Xiao et al., 2021*). The microbiota also differentiates from the environmental pool of microorganisms by neutral evolutionary forces, according to the proportions of the strains recruited, by the phenomenon called founding effect (*Robinson et al., 2018*).

Ecological succession does not follow the same patterns depending on the species studied. For example, it has been shown that alpha diversity (species richness) tends to increase in viviparous species whose microbiota is firstly transmitted at birth by the parents, and experiments in Derome laboratory are currently assessing whether the transmission of microbiota can occur during pregnancy (soon to be published), while it decreases in oviparous species after the first recruitment from environment (*Magnadóttir, 2006*; *Stephens et al., 2015a*; *Llewellyn et al., 2016*; *Sylvain & Derome, 2017*).

For zebrafish, species richness decreases over time, which could be explained by microbial competition within the bacterial community as well as the gradual exclusion of species by the immune system as it develops, the later being considered a potential ecological filter for the zebrafish microbiota (*Stagaman et al., 2017*). From zebrafish larval to adult stage, relative abundance within the microbiota of Fusobacteria increased, and Actinobacteria decreased, with no direct correlation with the bacteria detected in the type of food provided (*Stephens et al., 2016*). However it was also described that switching from live food to formulated food is associated to a switch of the dominant phylum (from Proteobacteria to Firmicutes) in the yellowtail kingfish (*Wilkes Walburn et al., 2018*).

A switch was also observed at 10 dpf in zebrafish larvae, with a reduction in microbiota richness, corresponding to the onset of larger prey foraging (*e.g.*, transition from rotifers to artemia in fish facilities), after which the relative abundance of the $\gamma$-Proteobacteria stop increasing and begin to decrease, while the opposite pattern is observed for $\beta$-Proteobacteria as represented in the summary Fig. 2 (*Stephens et al., 2015a*). In addition, the gut-microbial ecological succession of larval zebrafishes was not explained by the environment but

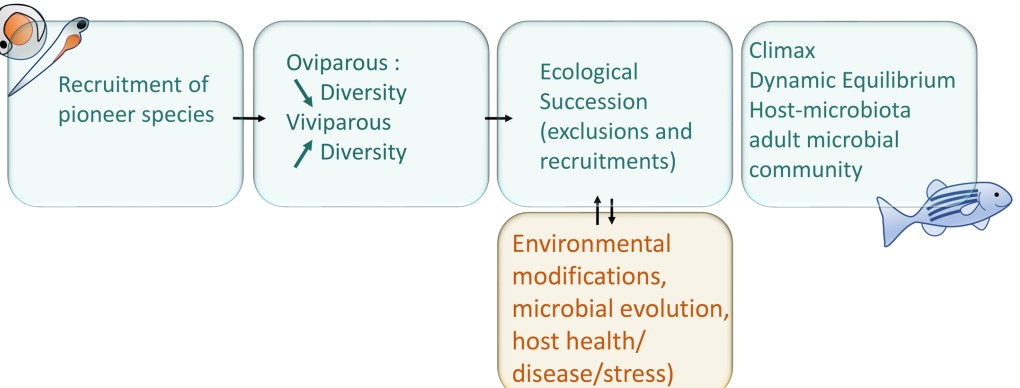

**Figure 1** Schematic representation of main interactions involved in the microbial succession during fish development.

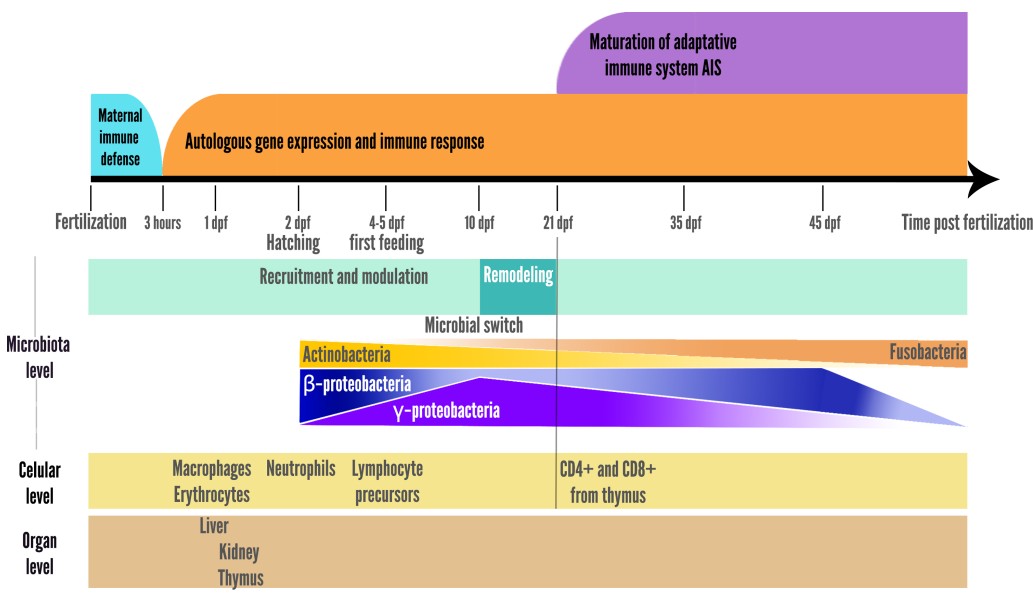

**Figure 2** Schematic representation of the two parallel setups in fish development: microbial community ecological succession and immune system maturation.

mainly by the effect of fish developmental stage and immunology (*Xiao et al., 2021*). From 35 dpf, the adaptive immune system is settled in the adult while gut microbiota richness decreases, then stabilizes from 75 dpf onwards (*Stephens et al., 2015a*; *De Bruijn et al., 2018*; *López Nadal et al., 2020*). This loss of richness from 35 dph translates into an accelerated taxonomic divergence between gut microbiota and environmental communities during fish development (*Stephens et al., 2015a*).

For the Atlantic salmon, Proteobacteria is the most abundant phylum at the embryonic stages (in this study, whole organisms were used for these early developmental stages as the embryos were not dissected for individuals before 7 weeks post hatching) similarly to the

observations in zebrafishes, and with mainly Betaproteobacteria at first stages, including some egg surface-specific species (*Lokesh et al., 2018*). Proteobacteria of the intestine microbiota then decreased, to only later regain dominance in the intestinal microbiota after transition to seawater (*Lokesh et al., 2018*). Species richness of the intestinal microbiota was described to decrease throughout the salmon life (between 20 to 44 wph) (*Lokesh et al., 2018*). A shift in skin (*Lokesh & Kiron, 2016*) and gut (*Llewellyn et al., 2016*) microbial composition was observed due to the migration from freshwater to seawater, which corresponds to the smoltification life stage (*Lokesh & Kiron, 2016*; *Llewellyn et al., 2016*).

As the adaptive immune system is being established, the microbiota is remodeled according to the bacteria encountered in the early stages. In fact, at the early stages the organism has been educated to tolerate certain bacterial strains that were not harmful. New strains recognized as harmful or ''non-self'' will trigger their exclusion by the immune system. Once the adaptive immune system gets full maturity, fish can maintain a relatively stable microbiota if the fish remain in a stable environment (*i.e.,* attaining the climax of the ecological succession), thus adequately protecting the host against pathogens (*Magnadóttir, 2006*).

*Co-evolved versus opportunist symbionts.* Co-evolved symbionts refer to microorganisms that have evolved in close association with a particular fish species. Some of these close associations can result in dependency, where the need for a certain product or function provided by one partner is provided by the other. This dependency can be observed, for instance, in the development of cell types or organs that could be impaired in absence of the needed symbiotic partner. For example, for developing a cell type or an organ. In zebrafish the absence of specific microbiota members able to produce the protein $\beta$ Cell Expansion Factor A (BefA) leads to an impaired development of pancreatic $\beta$ cells. This protein, sufficient and crucial to induce $\beta$ cell proliferation is not naturally produced by the fish itself, relying on the presence of the microbiota producers (*Hill et al., 2016*). This example vividly illustrates the zebrafish's dependency on particular microbiota members needed for the development of pancreatic cells. Another example of dependency is found in lantern fish having developed a specific organ (illicium) to host bioluminescent bacteria, the light emitting function of the organ depending on the presence of the bacterial strains which co-evolved with the fish host (*Freed et al., 2019*).

For salmonids, multiple studies reported a predominance of mycoplasmas in the gut microbiotas, demonstrating their strong association and co-evolution. For examples, an article estimated mycoplasmas to be 96% of the total microorganisms in the distal intestine of wild salmon (*Holben et al., 2002*), another study found *Mycoplasma sp.* To be the most frequent OTUs in Atlantic salmon digesta (*Abid et al., 2013*). A different study reported that *Mycoplasmataceae* are abundantly found in all life stages of Atlantic salmon (*Llewellyn et al., 2016*), and another study reported that gut microbiota of Atlantic salmon is dominated by mycoplasma (*Jin et al., 2019*). A very recent article has firmly established this co-evolution more explicitly by demonstrating that the coexistence of *Mycoplasma* (once again identified as the dominant species in the gut microbiota) with salmon was also linked to a co-diversification of the two partners (*Rasmussen et al., 2023*).

In the literature, it has been described that for adult fishes to possess a healthy microbiota, with most bacterial species having a positive or neutral effect on the host and that co-evolved with the fish species in a positive interaction, a continuous ecological succession is required during early life. Indeed, the repeated perturbations like disinfections or antibiotics utilization during the succession steps disrupt the recruited community in a non-specific way (*Vadstein et al., 2018*). By interrupting the gradual selection of beneficial microbial strains, repeated perturbations lead to the shaping of an unbalanced microbiota, which can no longer exert resistance to colonization of opportunistic and/or pathogenic strains, thus increasing disease susceptibility (*Kim, Covington & Pamer, 2017*; *Vadstein et al., 2018*).

*Internal host factors influencing the microbiota.* The action of the innate immune system and the intrinsic properties of the host microenvironment favoring the establishment of microorganisms could explain the repetitive observation of similar or identical microbial species associated with the same host species (called the core microbiota). This core microbiota is associated to the same fish species despite the rearing in different environments, as it was published in zebrafish, either from wild or captive environments, and more strikingly from freshwater and saltwater Atlantic salmons (*Roeselers et al., 2011*; *Rudi et al., 2018*). Thus, the host genotype drives the recruitment and maintenance of its microbial core symbionts (*Boutin et al., 2014*; *Sylvain et al., 2022b*).

Fish factors influencing the microbiota include the genetics (fish species, including specificities of its physiology and immune system), sex and age (*Xia et al., 2022*).

The host genetics and associated physiology account for the variations in intestinal microbiota observed among species. For example, experiments involving the inoculation of zebrafish with the intestinal microbiota of mice resulted in the microbiota switching to increase the microorganisms of the Proteobacteria phylum, thereby resembling the zebrafish's own microbiota (*Rawls et al., 2006*). The microbiota composition changes throughout the development and life stages of fish (*Stephens et al., 2016*), for zebrafish for example juveniles were described to have higher richness in intestinal microbiota than older individuals (*Cantas et al., 2012*). Utilization of zebrafish mutants highlights the role of host genes involved in the immune system in shaping the microbiota, as a mutant gene depletion induces a shift in microbiota composition. For example, RAG 1 depleted zebrafish lacking the adaptive immune system display a lower $\beta$-diversity than the wild-type zebrafish (*Stagaman et al., 2017*).

*Environmental factors influencing the microbiota.* In Atlantic salmon, apart from the stable core microbiota, artificially raised fish also exhibit a distinct global microbiota from wild fish and demonstrate a lower survival rate when reintroduced in wild environments, highlighting that not only the host, but also the environment shapes the microbiota (*Lavoie et al., 2018*; *Lavoie et al., 2021*). This environmental effect on non-core microorganisms is also observed in zebrafish raised in wild environments compared to laboratory-raised zebrafish (*Roeselers et al., 2011*). This highlights that the pioneer microbial community colonizing fish during the very first steps of immune system ontogeny leaves a permanent imprint on later fish and microbiota stages, which in turn determines fish adult fitness

(*Lavoie et al., 2021*). To test the effect of variations in microbial colonization during the early life stages on microbial assemblages in the adult stages, researchers placed wild juveniles on the one hand, and captive-bred juveniles on the other, in three experimental environments (standard rearing conditions, rearing conditions with an enriched diet and simulated wild conditions) shared by the two types of juvenile, for 6 weeks (*Uren Webster et al., 2020*). Faecal and cutaneous microbiome of each fish was analyzed before and after transfer to the experimental environments and the researchers demonstrated the existence of historical early life colonization effects reflected in the transferred fish, including bacterial DNA sequences specific to captive conditions. This demonstrated how environmental conditions experienced during early life, and particularly the pioneer microbiota composition, can have a long-term influence on the host microbiome and its health (*Alberdi et al., 2016*; *Uren Webster et al., 2020*).

The environmental factors influencing the microbiota composition are the salinity with a high importance, temperature, pH, antibiotics, chemicals found in the water and water composition, as well as molecules synthetized by surrounding individuals (in fish schools and also by other animals) (*Sylvain et al., 2016*; *Li et al., 2022*). Salinity is a key factor in shaping initial microbial communities in water (*Wang et al., 2012b*), and also of the fish microbiota. For the Atlantic salmon, a transfer in saltwater induces a modification of the intestinal and cutaneous microbiota (*Jaramillo-Torres et al., 2019*). The significant impact of saltwater in regulating the intestinal microbiota of Atlantic salmon was also demonstrated by other teams (inducing a reduction in the presence of various LAB genera) (*Jaramillo-Torres et al., 2019*; *Dehler, Secombes & Martin, 2017*). The pH has also been described to have a significant impact on fish microbiota for fish species capable of withstanding substantial fluctuations in pH levels within its native habitat (*Sylvain et al., 2016*). The temperature also affects microbiota in composition, and microbiota functions (effects on fish health). For example, a lower temperature can enable *Aeromonas salmonicida* to exert a pathogenic activity while a higher temperature attenuates the virulence, resulting in a strain harmless for the fish (*Ishiguro et al., 1981*; *Li et al., 2022*). For chum salmon, it was reported that an increased or reduced temperature was able to cause a switch in fecal microbiota associated to an increased detection of opportunistic pathogens (*Ghosh et al., 2022*). Finally, the use of antibiotics to treat bacterial infections, and their release into the environment, can disrupt the fish microbiota, potentially leading to the promotion of opportunistic microorganisms within the fish's microbial community (*Vadstein et al., 2018*).

*Intramicrobiota interactions influencing the microbiota.*

**Positive associations:** Microbial network analysis has highlighted that some bacterial strains favour the recruitment and growth of other specific strains, therefore strongly influencing the microbial community diversity (*Kehe et al., 2021*). These network topologies can result either from mutualism, syntrophy (*i.e.,* metabolic dependence, the interaction of obligate mutualistic organisms combining their metabolic abilities to metabolize an element that cannot be processed by any of the organisms individually), or metabolic redundancy (*i.e.,* foraging on the same resource) (*Morris, Lenski & Zinser, 2012*;

*Weiss et al., 2016*). In addition, some bacteria can generate modules of close microbe-to-microbe interactions with some other strains, and thus constituting core microbiomes with positive impacts on the host (*Zelezniak et al., 2015*; *Yajima et al., 2022*). For example, in Atlantic salmon parr and carp adults, gut microbial community networks are dominated by positive interactions, denoting mutualism or synergy between bacterial species (*Khurana et al., 2021*; *Lavoie et al., 2021*).

**Negative interactions:** In the microbial community, each strain or group of strains sharing niche specializations struggles for survival and to keep occupying its ecological niche (*Jackson et al., 2018*). In fish gut, the bacterial composition and species abundance varies depending on the gut sections considered (*Nielsen et al., 2017*). Competitive exclusion between bacteria is made possible thanks to the production of various antimicrobial molecules (such as bacteriocins: antimicrobial compounds produced by bacteria and fighting against the other bacteria) (*Melo-Bolívar et al., 2019*), and also happens between two bacterial species requiring the same nutritional resources (*Hibbing et al., 2010*; *Mullineaux-Sanders et al., 2018*). Bacteria can also produce anti-inflammatory compounds in order to prevent host immune response and thus securing their ecological niche (provided by the healthy host and the nutrients shared with it) (*López Nadal et al., 2020*; *Jenab, Roghanian & Emtiazi, 2020*). In addition, the production of anti-inflammatory compounds could benefit nearby bacteria (then also tolerated by the host). It is interesting to note that mutual exclusion of bacteria sometimes involves tightly phylogenetically related species, potentially because they compete for the same resources: in the Flavobacteriales, individual strains of a commensal *Flavobacterium sp.* and of *Chryseobacterium massiliae* ensures protection against *F. columnare* infection in the rainbow trout (*Pérez-Pascual et al., 2021*). *Aeromonas sobria* TM18 strain was demonstrated to strongly inhibit *Aeromonas salmonicida* subsp. *Salmonicida* in Brook trout (*Gauthier et al., 2019*). *Lactobacillus plantarum* CLFP 238, and *Lactobacillus mesenteroides* CLFP 196, when administered as probiotics to the rainbow trout, exert a competitive exclusion of pathogenic strain *Lactobacillus garvieae* (*Vendrell et al., 2008*). These probiotics strains were naturally co-occuring in gut with their pathogenic relatives.

Furthermore, fungi are also involved in fish host defense as they can inhibit pathogenic bacteria. Colonization of zebrafish larvae with multiple yeast strains resulted in an enhanced immune response against the pathogen *Vibrio anguillarum* and a better survival rate relatively to control group (*Caruffo et al., 2015*; *Caruffo et al., 2016*), therefore, highlighting the promising fungi probiotics formulations.

Contrariwise, it was described that some bacteria isolated from the microbiota can exert antifungal effects, protecting the fish host from fungal pathogens. For example, isolation and *in vitro* culture of rainbow trout microbiota strains (*Arthrobacter sp., Psychrobacter sp.*) and combined skin aerobic bacterial samples exerted strong inhibitory effects on two recurrent fish pathogens: *Saprolegnia australis* and *Mucor hiemalis*. These findings emphasize the beneficial function of fish symbiotic bacterial communities, and the promise of their use towards developing a sustainable managing of aquatic fungal diseases (*Lowrey et al., 2015*).

**Resistance to colonization:** The presence of a previously established bacterial community can influence population bottlenecks through a process called resistance

to colonization. This phenomenon is observed as hosts possess a diverse stable microbiota, which helps prevent the colonization by a pathogen (*Buffie & Pamer, 2013*; *Stephens et al., 2015b*). The resistance can be due to the presence of a bacteria inhibiting the pathogen or by the occupation of the resources by a bacterial community. For example, it was demonstrated that in the zebrafish, an individual bacterial strain, *Chryseobacterium massiliae*, is able to protect from a *Flavobacterium columnare* infection, but also that a synthetic community assembly of nine strains isolated from zebrafish are also able to protect against *F. columnare* despite the fact that none of these strains individually protects from the pathogen, demonstrating community-level resistance (*Stressmann et al., 2021*). Similarly, in the rainbow trout, axenic larvae are more susceptible to *F. columnare* than larvae having a conventional microbiota, and the addition of 11 bacterial strains to axenic larvae was sufficient to restore the fish's protection from *F. columnare* infection (*Pérez-Pascual et al., 2021*).

## Ontogeny of the interaction between immune system and microbiota
### *Mature immune system/microbiota interaction*

In healthy adult fish, microbiota constantly interacts with the immune system. The bacterial cells can interact with the immune cells, be sensed by dendritic cells, phagocytosed and presented by antigen-presenting cells (professional CPAs such as macrophages, dendritic cells, B cells, and nonprofessional CPAs like thymic epithelial cells and vascular endothelial cells). The bacteria also interact with the host's molecules (such as antibodies, antimicrobial peptides). What triggers an immune response is not only the presence of foreign/non-self-components, but the global and local molecular context: the pro-inflammatory or anti-inflammatory environment. In some circumstances, bacteria secreting pro-inflammatory molecules can prompt an immune rejection as the inflammation triggers an exclusion reaction from the immune system. Because microbiota and immune system are in a dynamic equilibrium, influencing constantly each other (Fig. 3), it is relevant to consider the theory of discontinuity recently developed in Immunology as a theoretical framework (*Pradeu & Vivier, 2016*). To that respect, it seems pointless, even useless to look for which of the two systems is the cause of the disease: *e.g.*, is it the depleted microbiota no more exerting its anti-inflammatory effect nor colonization resistance, leading to inflammation driven diseases or is it the inflammation itself causing the microbial dysbiosis? The theory of discontinuity hypothesizes that the discontinuity in the microbial succession process and/or in the development of the host immune system is a pervasive element preceding the onset of the illness.

*Homeostasis of the immune system and microbiota taxonomic composition.* Microorganisms conserved and tolerated by the fish have co-evolved with it over a long period of time, suggesting that the phenotypic outcomes of symbiosis between a fish and a specific microbiota were tightly selected. This supports the theory that the holobiont (comprising the fish and its microbiota) is an evolutionary unit selected through evolutionary times (*Margulis, 1991*; *Baedke, Fábregas-Tejeda & Nieves Delgado, 2020*). Indeed, the phylosymbiosis, which describes the co-evolution of host organisms and

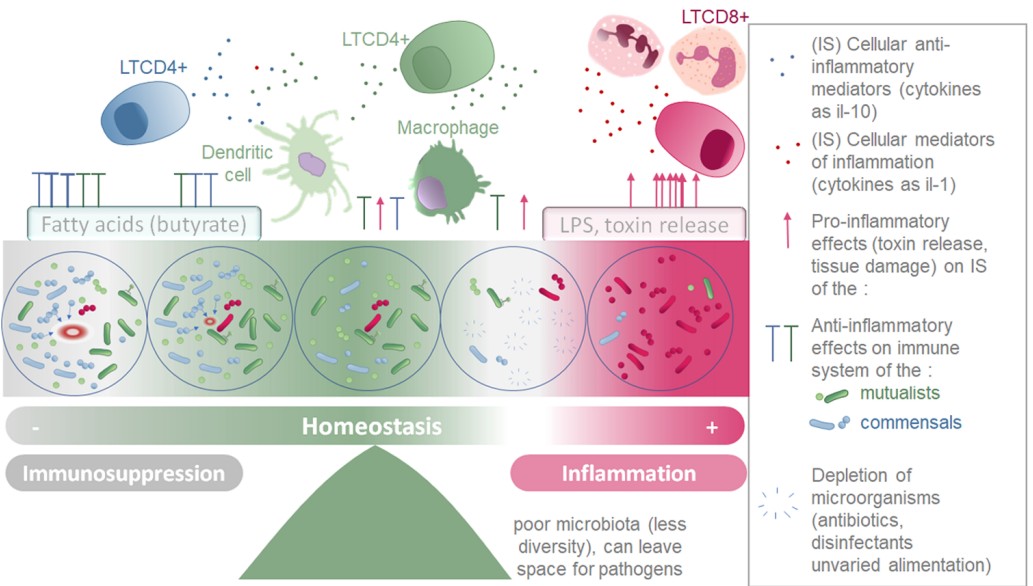

**Figure 3** Schematic representation of the two parallel equilibriums: immune system (inflammation and immunosuppression) and microbiota composition equilibrium involved in maintaining healthy fish. Immune system (SI) balance depending on immune modulation of cellular response and on the microbiota associated to the host at each microenvironment considered (gills, intestine, skin).

their core microbiota, can be successfully detected for many fish species (currently 44 species described) (*Chiarello et al., 2018*; *Mazel et al., 2018*; *Lim & Bordenstein, 2020*). These co-evolved microorganism communities do not trigger an excessive inflammatory reaction in healthy individuals, but instead are associated to an immune equilibrium enabling their survival with the fish (*Pérez et al., 2010*). Symbiotic microorganisms have the ability to produce an extensive variety of metabolites, including tolerogenic, anti-inflammatory metabolites, pro-inflammatory molecules (especially in dysbiosis manifestations) antibiotics, pigments, vitamins, antioxidants and enzymes (*Dehhaghi, Kazemi Shariat Panahi & Guillemin, 2019*). In Table 4, we present a selection of metabolites that are secreted by microorganisms in microbial communities associated with hosts, and that have been described to affect the immune system locally, or in some cases, more globally by being transported through the blood stream. Although the mechanisms and precise metabolites involved are not always deciphered, numerous species of microbiota have been found to affect fish immune system (*Murdoch & Rawls, 2019*; *Ringøet al., 2022*). The species able to alter the host environment may have more chances to associate durably with the host, and help conservation of themselves and of the overall microbial community they belong to (*Mallott & Amato, 2021*). This equilibrium is represented by the central part of the Fig. 3.

A healthy host-microbiota interaction is a state of equilibrium in which bacteria and host regulate the molecular and cellular environment in order to be mutually beneficial. To avoid immune response, the microbiota must turn down production of pro-inflammatory molecules, and avoid inducing damage that would create damage-associated molecular

**Table 4  Selection of metabolites influencing the host immune response.** Sources: (a) *Fu et al., 2021*; (b) *Roager & Licht, 2018*; *Ma, Liu & Wang, 2022*; (c) *O'Mahony et al., 2015*; (d) *Haque et al., 2022*; (e) *Holzer, Reichmann & Farzi, 2012*; *Louis, Hold & Flint, 2014*; (f) *Ahmadifar et al., 2021*; *Ahmadi et al., 2022*; (g) *Schneider et al., 2016*.

| Metabolite | Bacterial genera producer | Effect on immune system | References |
|---|---|---|---|
| Tryptophan metabolites | *Lactococcus, Lactobacillus, Streptococcus Escherichia coli,* and *Klebsiella* | Directly reduce inflammation in the liver or indirectly affect liver function by modulating the intestinal barrier, (released in bloodstream). Improves growth of triploid crucian carp, and intestinal morphology. (increases microbiota's richness and evenness) | (a) (b) (c) |
| SCFAs (Short Chain Fatty Acids) such as: Acetate Propionate Butyrate | | Maintain intestinal homeostasis and regulate fatty acid oxidation and inflammation. SCFAs from non-digestible carbohydrates participate in microbiota modulation, cellular differentiation, and inflammation reduction) Impacts brain function: influences enteroendocrine serotonin production and stimulates the release of peptide YY. Suppression of inflammation and cancer | (d), (e) (e) |
| Neurotransmitters Choline metabolites Tryptophan | | Released in bloodstream (gut-brain axis) | (d) |
| Polyphenols, phenolic acids | | (From phytochemicals) effect: xenobiotic detoxification, microbiota modulation, cellular differentiation, reduced apoptosis induced by ROS overproduction, dampening inflammation. | (f) |
| Polyamines | | Increase inflammation, ROS production, genotoxicity | (f) |
| Hydrogen sulfide | | Inflammation, ROS production, genotoxicity | (f) |
| Acetaldehyde | | (From ethanol) ROS production, genotoxicity | (f) |
| Acetaldehyde metabolizing components | *Lactobacillus rhamnosus GG* | Prevents the increase of pathogenic bacteria, protects the intestinal barrier from ethanol exposure (modulate microbiota) | (g) |
| Phytochemicals | | Anti-inflammatory effects. *In vitro* and *in vivo* models indicate that they inhibit pro-inflammatory mediators (TNF), IL-6 and prostanoids). Microorganism-associated molecular patterns (MAMPS) of some commensal bacteria are thought to contribute to anti-inflammatory signaling | (g) |

patterns (DAMPS) that, in turn, would activate host secretion of inflammatory cytokines. At the same time, the microbiota must maintain optimal taxonomic diversity and prevent any imbalance in terms of richness, evenness, and interaction patterns, to avoid diseases associated with microbial dysbiosis (*Cheaib et al., 2020*; *Pascal et al., 2021*) (Fig. 3). Indeed, a diversified microbiota is essential to the host health, possibly because mucosal antibodies have a preference for selectively binding to certain taxonomically distinct subsets among the commensal microbes (*van der Waaij et al., 1996*). Secretion of mucosal antibodies that attach to commensals prevents their passage through the host epithelial barrier, while allowing them to remain in close proximity to host surfaces and provide their beneficial functions (*Vaishnava & Hooper, 2007*). Improvement of host health mediated by gut microbiota was investigated in common carp, by characterizing both bacterial community taxonomic composition and host transcriptome. This study revealed that a performance enhancement was associated with an increase of alpha diversity index (chao1) and entropy measures (Shannon, Simpson). Interestingly, most of the carp transcripts

that were associated with performance enhancements were related to the immune system, bacterial community and cell differentiation categories (*Su et al., 2021*). This highlights that bacterial composition is associated to the growth performance and host immune system's gene expression and cell differentiation during fish development. In response to the healthy fish microbial community not harming and secreting anti-inflammatory compounds, immune systems are set up to reinforce microorganism tolerance. The initial microbiota plays a role in the immune stimulation and tolerance of larvae, and while it may not necessarily persist for long-term fish colonization, it might still impact the future immune responses of adult fish colonized at early stages (*Gomez, Sunyer & Salinas, 2013*; *Salinas, 2015*). Indeed, paradoxically, the most abundant type of bacteria in the community (gram-negative) harbours highly inflammatory LPS ($\gamma$-proteobacteria, $\beta$-proteobacteria, Fusobacteria). LPS activates inflammation, neutrophil recruitment, and the expression of inflammatory cytokines through the MyD88 signaling pathway (*Wang et al., 2014*). To cope with these inflammatory compounds, the host evolved to express and secrete intestinal alkaline phosphatase (IAP) which detoxifies the LPS by dephosphorylation. By preventing inflammation, IAP exerts an essential role in tolerance of the microbiota members (*Bates et al., 2006*; *Vaishnava & Hooper, 2007*; *Estaki, De Coffe & Gibson, 2014*). Overall, equilibrium between LPS presenting bacteria and IAP production enables the preservation of homeostasis.

*Discontinuity of the host-microbiota homeostasis and immunosuppression.* A discontinuity of homeostasis leading to immunosuppression, potentially facilitating opportunistic infections, is shown in the left side of Fig. 3. An immune system impairment causing excessive tolerance makes the host susceptible to opportunistic pathogens, especially if the microbiota is depleted. Indeed, neither host (by its own defenses) nor microbiota (through competitive exclusion or resistance to colonization) can neutralize the pathogen. For instance, zebrafish depleted for TNF receptor 1 were more susceptible to the pathogen *M. abscessus*, as the depletion of signaling pathways, normally activated by the TNF, greatly limited the neutrophils and macrophage recruitment, enabling unlimited growth of the pathogen (*Bernut et al., 2016*; *Torraca & Mostowy, 2018*). Another study on zebrafish demonstrated that the bacteriostatic antibiotic Enrofloxacin (ENR) has an immunosuppressive effect and a general immunotoxicity, in addition to alter fish intestinal microbiota (*Qiu et al., 2022*). This study highlighted a significant correlation between gut community disturbances and global immunosuppressive responses induced by ENR exposure in fish, demonstrating that perturbing the microbiota can lead to immunosuppression (*Qiu et al., 2022*).

Environmental conditions can also directly or indirectly impact the fish and its interaction with microbiota. Some environmental stressors, such as changes in temperature (*Rosado et al., 2021*; *Ghosh et al., 2022*), oxygen levels (*Sylvain et al., 2019*; *Krotman et al., 2020*), exposure to pollutants such as cadmium, glyphosate for example (*Cheaib et al., 2020*; *Cheaib et al., 2021*; *Bellec et al., 2022*), or abrasive forces (due to manipulations in aquaculture, high fish concentrations favouring fish aggressive behaviours, nets utilization, or even lesions due to macroparasites such as *Lepeophtheirus salmonis*) impact the

microbiota (*Llewellyn et al., 2017*). Indeed, they lead to weakened first-line defenses in the body, such as mucus thickness reduction, skin wounds (compromising the integrity of the skin and mucus layers enables uncontrolled bacterial entry in the fish body), lessen production of microbiota synthetized antimicrobial compounds or disruption of the synthesis pathways of immune molecules. This allows opportunistic bacteria to colonize and invade host tissues (*Hansen & Olafsen, 1999*).

*Discontinuity of the host-microbiota homeostasis and inflammation.* Fish microbiota disruption triggered by antibiotics or disinfectants favours intestinal colonization and spread of fast-growing microorganisms, often consisting in opportunistic pathogens on one hand (*Vadstein et al., 2018*), and, on the other hand, local or systemic inflammation potentially increasing susceptibility to infection and mortality (*Robledo et al., 2016*), as represented in the right side of Fig. 3.

Antibiotic-induced disruption of the microbiota can liberate ecological niches, suppress the above-mentioned mechanisms of competitive exclusion and colonization resistance, and thus create conditions promoting the overgrowth of opportunistic pathogens, which in turn can increase the risk of systemic infection (*Taur & Pamer, 2013*; *Legrand et al., 2020*). In aquaculture, disinfections are routinely performed to get rid of pathogens. However, by eliminating bacterial species indiscriminately, disinfection removes many beneficial K-strategists (slow-growing-species, specialized to their environment, the fish, like oligotrophs). Then, as the ecological niches previously occupied by K-strategists are made available, this strategy has a boomerang effect by favouring in turn fast-growing generalist bacteria, also called r-strategists (bacteria less specialized to a specific environment, like copiotrophs) including opportunistic pathogens, such as *Vibrio* species (*Hansen & Olafsen, 1999*; *Vadstein et al., 2018*).

### Ontogeny of the immune system/microbiota interaction

*A first inherited immune defense –first microbial filter.* At very early stages of development, the fish immune system is not yet developed (no immune cells), but the fish egg or larvae are not completely defenseless. In oviparous species, the chorion, the first mechanical protection, contains the perivitelline fluid in which the fertilized zygote lies. This fluid, transmitted vertically during oviposition, provides an initial defense against external pathogens, and acts as a first filter for colonizing microbes. To this respect, perivitelline fluid contains maternal antibodies conferring a protection against infections to which the parent has been exposed, as well as lysozyme, intelectins, proteins of the complement system, and even antibodies from the mother (*Kanlis et al., 1995*; *Løvoll et al., 2006*; *Mulero et al., 2007*; *Wang et al., 2008*; *Wang et al., 2009*; *Wang et al., 2012a*; *Wang et al., 2012b*; *Li et al., 2011*; *De la Paz et al., 2020*). Therefore, egg possesses an immune defense with components of both the innate and adaptive maternal systems. The mother can also transmit hormones to her offspring, which, depending on her own stress level, will influence the early transcription of immune genes in the larvae (*Sopinka et al., 2017*). Based on these observations, experiments have been conducted to improve maternal immunity in order to increase offspring survival (*Hanif, Bakopoulos & Dimitriadis, 2004*; *Wang &*

*Zhang, 2010*). For example, effective vaccination of parents during egg development in sea bream lend the eggs an improved first defense, indicated by a higher humoral response (antibodies against the agent targeted by the vaccination, lysozyme activity) supposedly transferred by the mother (*Hanif, Bakopoulos & Dimitriadis, 2004*). In another study, it was demonstrated that the lysozyme and complement transferred by the mother have bacteriolytic activity to protect the eggs. This suggests that practicing vaccinations before egg laying could be a sound strategy to explore, demanding less resources and investments from farmers.

Before hatching, the eggs can be susceptible to infections, for example *Saprolegnia parasitica*, an endemic fungal parasite in fresh water habitats worldwide, causes severe losses at the egg stage in aquaculture (*Earle & Hintz, 2014*). To prevent the onset of diseases, prophylactic disinfection and antibiotic administration are widely used in aquaculture (*Subramani & Michael, 2017*; *USDA, 2017*). However, in addition to posing a risk to fish development (*Kent et al., 2014*; *Ren et al., 2022*), antibiotics damage the microbiota associated with the eggs or larvae, disrupting the microbiota ontogeny, directly impacting the fish immune system development and enabling r-strategists (including opportunistic pathogens) to colonize the eggs after disinfection processes (*Bates et al., 2006*; *Ministry of Natural Resources, 2009*; *Kent et al., 2014*; *Vadstein et al., 2018*).

At hatching, the environmental bacteria are suddenly put in direct contact with the fish mucosal surfaces, and the host-microbiota dialog begins. From the diversified environmental microbial pool, only species able to cope with the host local physical and chemical properties—mucus layer composition and first maternal and autologous innate immune response—will successfully colonize the fish body surfaces, therefore composing the pioneering microbial community (*Kelly & Salinas, 2017*). Most fish species studied do not provide parental care or undergo live birth (*i.e.,* viviparous and viviparous-like species) that would result in a more species specific microbiota at birth, but there are a few exceptions for which the parents have an important role in the early microbiota establishment, such as the discus larvae feeding on skin-mucus from the genitors (*Sylvain & Derome, 2017*) or pipefish incubating pouch (*Beemelmanns et al., 2019*).

At the larval stage, depletion of the whole microbiota due to disinfection can suppress the protective microbiota, thus rendering fish susceptible to infection by opportunistic pathogens like *F. columnare* (*Pérez-Pascual et al., 2021*). Unfortunately, most studies on egg and larvae disinfection did not mention the precise developmental stage at which the treatment was applied, thus rendering comparison of one study to another tedious. For instance, it was reported that disinfecting cod eggs with Hydrogen peroxide resulted in better hatching rates for the more developed eggs, but not for the early stages of egg development (*Peck et al., 2004*; *De Swaef et al., 2016*). On the opposite, rainbow trout embryos closer to the eyed stage were more susceptible to disinfection with hydrogen peroxide than earlier embryos just after fertilization (*Wagner et al., 2010*). All these observations, although not standardized in terms of developmental stage from one species to another, highlight the importance of both timing and fish developmental stage at which a disinfection treatment is applied. Both of these parameters deserve further studies in order to draw conclusions about the balance between beneficial and adverse effects of disinfection protocols.

As soon as the microbiota becomes established, microorganisms interact between each other and with the host. By consuming and metabolizing host derived resources, stimulating the immune system and interacting with the nervous system, this rich microbial community gradually modifies its local environment, thereby influencing the host development and the differentiation of its immune system (*Bates et al., 2006*), including by inducing epigenetic changes (*Gerhauser, 2018*), and small RNA (*Hu et al., 2017*; *Raza et al., 2022*). For instance, the secretion of fish mucus is stimulated by the exposition to microorganisms (*van der Marel et al., 2010*). Then, the presence of bacteria influences the immune system activation as it has been demonstrated that the activation of TLRs by bacterial LPS activates the MyD88 pathways which in return lead to the expression of inflammatory cytokines and intestinal alkaline phosphatases (IAP) that will reduce the inflammation and LPS toxicity (see previous section). Reduced expression of IAP in germ-free zebrafishes, and returns to control levels in germ-free animals upon microbiota colonization confirms the crucial role of bacteria in educating host immune system (*Bates et al., 2007*). At the larval stage, microbiota depletion due to disinfection can suppress beneficial microbial strains, thus increasing susceptibility to infections by opportunistic pathogens like *F. columnare* (*Pérez-Pascual et al., 2021*). When the fish switches to first feeding (at 5 to 10dpf for zebrafish), the food serves as an entry route for new microbial strains, thus leading a restructuration of the microbiota. This step is associated to a decrease in microbial diversity for zebrafish (*Ingerslev et al., 2014*; *Wilkes Walburn et al., 2018*; *López Nadal et al., 2020*), likely resulting from the preferential selection of health-promoting K-strategist microorganisms within the fish host. For anadromous species, a switch of microbiota and inflammation status accompanies the transfer from freshwater to saltwater: the immune response is silenced and both skin and gut microbiota composition change, as observed in Atlantic salmon (*Lokesh & Kiron, 2016*; *Llewellyn et al., 2016*; *Dehler, Secombes & Martin, 2017*) and Artic char (*Ojima et al., 2009*; *Hamilton et al., 2019*). Finally, at the adult stage, a relative stable equilibrium between the microbiota and the fish immune system enables fish to host the beneficial symbionts while eliminating harmful ones, therefore ensuring optimal health. The impact of a maturing immune system on microbiota composition in fish still needs to be further investigated.

## CONCLUSIONS

Typically, the microbiota and the immune system are studied as two separate interacting entities. Because of the growing attention to the holobiont theory (*Baedke, Fábregas-Tejeda & Nieves Delgado, 2020*), more and more researchers are apprehending the interaction between the microbiota and host immune system as a single system responding to evolutionary forces. In particular, the host fish is now considered as an ecosystem providing food, energy and protection to its own cells and associated microbial cells (microbiota), allowing for the ecological succession of microbiota and immune cells (*Sevellec, Derome & Bernatchez, 2018*; *Simon et al., 2019*; *Sylvain et al., 2022a*). Immune system development and maturation is a continuous process which is controlled by many factors such as environmental parameters, parental immunology, organ development and dialogues

 

between microbial communities and fish immune system. These factors are constantly changing, allowing the immune system to be reactive against pathogens and allowing beneficial commensal microorganisms to thrive in association with the fish host. The holobiont phenotype and fitness are shaped by its developmental history resulting from interactions with the constantly changing microbiota and immune system.

While the host immune system constantly evolves in response to resident microbial communities, the immune response adapts to maintain up-to-date biological defenses according to all living forms encountered. Each provides new immune information that is taken into account to develop tailored responses in real time throughout fish development.

## FUTURE PERSPECTIVES

Due to the considerable variations in size, metabolic rate, dietary preferences, and habitat (*e.g.*, saltwater or freshwater) among different fish species, it is unsurprising to observe significant differences in their microbiota composition, despite sharing numerous anatomical, histological, and physiological features. Although zebrafish model has been extensively used in medicine to study human diseases, often by using gnotobiotic fish colonized with human microbiota (*Neely, 2017*; *Gomes & Mostowy, 2020*; *Lu et al., 2021*), researchers only recently considered comparing zebrafish with other fish species (*Jørgensen, 2020*). It would be interesting to investigate further whether this model can effectively simulate host immune-system and microbiota interactions occurring in other fish species.

Thus, a better understanding of interactions between immune-system and the microbiota could help to solve aquaculture industry challenges (*e.g.*, early larvae mortality) and to enable a better use of animal models in medical experiments such as zebrafish. Indeed, microbial community management is a promising strategy to strengthen fish responses to immunological challenges, which in turn will improve fish performance in a sustainable manner, *e.g.*, without the drawback of increased antibiotic resistance in fish and environment. Alternatives to the use of antibiotics and disinfections include the implementation of water recirculation systems (RAS) that promote the recruitment and maintenance of beneficial microorganisms, and the use of probiotics and phages (phage therapy), all of which are effective ways of preventing infections without impairing or creating discontinuities within the host-microbiota system.

We could conceive aquaculture without disinfection treatments which can also reduce or block the transmission of symbionts between parents and offspring and thus can reduce fitness and defenses by altering the immune system and microbiota (from the egg stage for oviparous species).

The microbiota and immune system studies collect large and complex datasets of diverse microbiotas in various contexts and environments, and it can be noted that artificial intelligence advances and techniques like machine learning might play a crucial role in analyzing microbiota in the future by leveraging its capabilities in integrating complex datasets, pattern recognition, and predictive modeling, to improve comprehension of associations between hosts and their microbiotas (*Giuffrè, Moretti & Tiribelli, 2023*).

## ACKNOWLEDGEMENTS

I would like to express my sincere gratitude to Dr. Steve Charette for his valuable advice. Additionally, I would also like to thank Dr. Claire Léveillé for providing me with valuable references on immunology and for taking the time to explain me some theoretical aspects. I would like to thank Felipe Andrés Cerpa Aguila for his writing advice.

### Funding

The authors received no funding for this work. The APC for this work was funded by the Government of Canada through Genome Canada and the Ontario Genomics Institute (OGI-184). The funders had no role in study design, data collection and analysis, decision to publish, or preparation of the manuscript.

### Grant Disclosures

The following grant information was disclosed by the authors:
Government of Canada, The Ontario Genomics Institute:  OGI-184.

### Competing Interests

The authors declare there are no competing interests.

### Author Contributions

- Lisa Zoé Auclert conceived and designed the experiments, performed the experiments, analyzed the data, prepared figures and/or tables, authored or reviewed drafts of the article, and approved the final draft.
- Mousumi Sarker Chhanda analyzed the data, authored or reviewed drafts of the article, and approved the final draft.
- Nicolas Derome analyzed the data, authored or reviewed drafts of the article, and approved the final draft.

### Data Availability

 This is a literature review.

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
