# Peer review of "Interwoven processes in fish development: microbial community succession and immune maturation"

_PeerJ, doi:10.7717/peerj.17051_

## Round 0.1 · original submission · Major Revisions

The revision should be improved to better attract the reader's interest. The first part refers only to immunity, it should be shortened and be seen to be necessary to understand the other part. Otherwise, the first part would not be necessary.
Many more suggestions have been provided by the reviewers..

**Language Note:** The review process has identified that the English language must be improved. PeerJ can provide language editing services - please contact us at [email protected] for pricing (be sure to provide your manuscript number and title). Alternatively, you should make your own arrangements to improve the language quality and provide details in your response letter. – PeerJ Staff

Reviewer 1 ·

Basic reporting

Clear, unambiguous, professional English language used throughout.
The document's English language requires significant improvement throughout.
Intro & background to show context. Literature well referenced & relevant.
The introduction and background lack proper contextualization of the topics under discussion, necessitating improvement.
Structure conforms to PeerJ standards, discipline norm, or improved for clarity.
Yes
Is the review of broad and cross-disciplinary interest and within the scope of the journal?
Yes
Has the field been reviewed recently? If so, is there a good reason for this review (different point of view, accessible to a different audience, etc.)?
The present review has distinct objectives when compared with other recent reviews on the same subject.
Does the Introduction adequately introduce the subject and make it clear who the audience is/what the motivation is?
Yes

Experimental design

Article content is within the Aims and Scope of the journal.
Yes
Rigorous investigation performed to a high technical & ethical standard.
Yes
Methods described with sufficient detail & information to replicate.
Not applicable.
Is the Survey Methodology consistent with a comprehensive, unbiased coverage of the subject? If not, what is missing?
Satisfactory
Are sources adequately cited? Quoted or paraphrased as appropriate?
The coverage of references lacks depth.
Is the review organized logically into coherent paragraphs/subsections?
The organization of the review lacks logical structure and cohesiveness. Numerous sections contain superficial discussions that contribute little value to the article's content

Validity of the findings

Impact and novelty not assessed. Meaningful replication encouraged where rationale & benefit to literature is clearly stated.
Not applicable
Conclusions are well stated, linked to original research question & limited to supporting results.
Yes
Is there a well developed and supported argument that meets the goals set out in the Introduction?
Moderately
Does the Conclusion identify unresolved questions / gaps / future directions?
Yes

Additional comments

The specific comments are given below.
Line 41-43: It is crucial to underscore the significance of vaccination in this context. Diseases affecting aquaculture necessitate a multifaceted approach beyond solely relying on antibiotics.
Line 72-75: It's important to clarify that the gut does not constitute the external body surface. The potential presence of microbes, including microbial DNA, within internal organs like the liver and blood should not be dismissed and requires consideration.
Line 78: Preceding reviews that have delved into analogous topics must be recognized. E.g. https://www.sciencedirect.com/science/article/pii/S1050464813007778
Line 87-89: The sentence needs rephrasing to convey a clearer message.
Line 93-94: The sentence lacks clarity.
Line 96-97: The sentence needs clarification.
Line 101: Could you clarify the term "setting"?
Line 108-109: This sentence appears irrelevant.
Line 141: The numbering of this section seems incorrect.
Lines 166-185: Although several of the points discussed here are "widely accepted", proper references are necessary. Additionally, it's essential to clarify whether these arrangements are observed in fish or if they are solely observed in mammals.
Line 141 I.A.a) Introduction on the immune system: This section lacks a comprehensive account of the fish immune system. The content seems heavily biased towards VDJ rearrangements. This section should encompass the fundamental functions of the immune system, its evolutionary significance, and accentuate species-wise differences and similarities, providing a holistic overview of the fish immune system. For instance, fish (cod and angler fish) lacking the adaptive arm of the immune system should be addressed, and the lungfish cocoon could be discussed to provide readers an overview of the extremes present in the fish defense system.
Line 198-199: The sentence's meaning is unclear. Capital letters are misplaced.
Line 218-219: The sentence lacks clarity.
Line 247: "LT," "LB," and "NK" should be written in full
Line 248: "CMH," "DAMPs," "PAMPs," and "TLRs" should be written in full as for their first mention
Line 206: I.A.b) Components of the immune system in adult teleost fishes: The level of detail provided for various tissues does not maintain consistent patterns. For instance, while the kidney and spleen receive coverage with regards to structural and cellular aspects, the discussion of the thymus centers predominantly around involution. To enhance the coherence of this section, a systematic approach should be adopted. This involves presenting the structure, cell composition, and molecular elements followed by specific details unique to each tissue.
Line 270, 287, and 302: Sections "Cellular Innate Immunity," "Cellular Adaptive Immunity," and "Molecular Immunity": These sections currently lack cohesion, consistency, and completion. It's unclear what the authors intend to convey. Notably, studies investigating the interactions between neutrophils and the host microbiome in zebrafish have been disregarded. If the objective of these sections is to components of immune system, a table format might be preferable, detailing cell types, functions, and associated species. Moreover, the content should be presented in a contextual manner, adhering to a clear framework.
Line 385-386: The sentence's clarity is compromised.
Line 395: Prior to exploring cell ontogeny, discussing the ontogeny of organs is more appropriate.
Line 395: I.B.b) Cellular Innate Immunity and Immune Organ Ontogeny: This section is incomplete and lacks appropriate referencing. Ontogeny of immune organs and cells is extensively researched across various teleosts such as carps, salmonids, and zebrafish. Presently, the section predominantly focuses on a few zebrafish studies. To enhance this section, inclusion of the appearance, maturation of immune cells, and concepts related to the recognition of commensals and pathogens is necessary.
Line 453: I.B.d) Immune System Influenced by Environmental Stress: The current state of this section reflects inadequate review. The section's title is misleading, as the effects of environmental stress encompass a broader scope with substantial literature available. The existing content under this section briefly mentions a couple of studies on the impact of cortisol on the immune system. Should this section be retained, an extensive expansion is warranted, either focusing on broader environmental effects or delving into the impacts of cortisol.
Lines 476-478: The sentence contains a slight misrepresentation. The statement about bacterial populations being the most diverse and abundant groups in fish requires nuanced phrasing since the literature lacks information regarding fish-associated fungi and viruses. Moreover, it's pertinent to note that bacterial diversity is low in adult salmonids, as evidenced by recent studies by Rasmussen et al. A relevant query arises regarding the dominance of other groups in this context. Notably, a recent preprint highlights diverse fungal phylotypes in salmon, which adds contextual depth to this discussion.
Line 485: Salinity is a decisive factor influencing community profiles. Numerous studies in teleosts support this notion and merit discussion.
Lines 511-512: The application of metagenomics in fish microbiome research remains extremely limited. This is attributed not only to its resource and time-intensive nature but also to challenges linked to sample collection and host DNA contamination. The constraints associated with metagenomics have been clearly demonstrated in the case of salmonids, as shown by Rasmussen et al. This crucial aspect must be elaborated upon in this context.
Line 516: II.A.b) Axenic Models to Understand Microbiota Functions: Within this section, it is essential to explore the notable distinctions in methods employed for establishing axenic models in fish and mammals. Moreover, a comprehensive examination of the limitations associated with such models is imperative. Delving into the maturation of the immune system and elucidating functional disparities between axenic fish and conventionally raised fish is necessary. Additionally, the inclusion of studies focusing on the axenic model of trout is warranted.
Line 539: II.A.c) The Microbiota as an Immune Organ System?: Presently, this section appears unfinished. A wealth of studies centered on zebrafish has elucidated the intricate interplay between the immune system and the commensal microbiota. It is paramount to incorporate discussions of these studies, such as the one mentioned here: (https://www.ncbi.nlm.nih.gov/pmc/articles/PMC5520148/).
Line 636-641: This section should specify the tissue to which the authors are referring.
Line 654: II.B.c) Co-evolved Versus Opportunist Symbionts: The discussion within this section remains insufficient. Notably, papers addressing the dependence of zebrafish on microbial proteins to facilitate the insulin response, as well as recent findings concerning salmonid mycoplasma, should be thoroughly explored in relation to the concept of co-evolution.
Line 666: II.B.d) Internal Host Factors Influencing the Microbiota: Currently, this section lacks substantial content and proper literature review.
Line 677: II.B.e) Environmental Factors Influencing the Microbiota: Within this section, a comprehensive overview detailing various environmental factors and their influence on the microbiota is necessary.
Line 890-892: These statements should be substantiated with appropriate references.
Line 923-924: The review's value could be greatly enhanced by integrating discussions of these experiments and their outcomes within the context of the current review.

Reviewer 2 ·

Basic reporting

The literature review thoroughly describes the fish immune system, the role of microbiota and the interaction of the two, explaining the implications for fish health and performance.
Overall I think the study provides an excellent perspective, it is very nicely and clearly written, and easy to follow.
I have only a few comments/suggestions for the authors:
- Title: it is a bit misleading. You talk about two intercorrelated setups and you present three. I understand it refers to microbiota and immunity, but as it is, the title is confusing a bit. So I propose to adjust it.
- Introduction: Please make it more clear (it is now hidden in the text), what is the point of this review. Why does this review differ from previous ones on fish immunity and microbiota (because there are several ones)?
- References: As mentioned the study is very nicely written and straightforward. However, it still remains a literature review. Therefore, attention must be paid to the long texts without a single reference. So I would strongly advise you to do a thorough reference check.
- The conclusions read more like another section. So I would advise concentrating on what the authors want to give as a final message. You could also split it further into future perspectives for example. But the section conclusions should really present conclusions and not further discussion.

Experimental design

The methodology is appropriate

Validity of the findings

Appropriately presented. However, a meta-analysis would have been more interesting to also show and compare the different findings, other than just reporting results.

Reviewer 3 ·

Basic reporting

The manuscript reviews the effect of microbiota on the immune system of fish. This type of review is very useful, especially for students or those who wish to know more about this topic's state of the art.

In its current form, the article reads more like two separate reviews with limited connection between them. The initial section extensively covers the immune system in fish, delving into intricate details such as molecular components, gene expression, and arrangements. However, this detailed exploration doesn't appear to have a clear link to the subsequent sections, particularly the discussion of the microbiota. For instance, from lines 161 to 186, the article mentions V(D)J genes, but it remains unclear how these genes relate to any specific modulation of the gene expression by microbial groups. To enhance the cohesion and relevance of the content, it may be beneficial to establish a more direct connection between the immune system description and its interaction with the microbiota.

The initial section, which describes the immune system, appears somewhat disjointed from the authors' primary focus, who intend to discuss the microbiota and its interaction with the immune system. The introductory part is excessively lengthy and occasionally challenging to comprehend due to its repetitive nature and long sentences. For instance, consider the sentence spanning lines 77 to 81; it would benefit from being divided into two separate sentences to enhance readability and comprehension.


My first recommendation would be to eliminate this first part or reduce it to the minimum necessary to understand or introduce the other two sections.

Another critical issue is the title; I consider it confusing. It presents two interrelated aspects of fish development and then lists three. It should be reworded.

While the article appears well-written, its overall style has room for improvement. I believe it is crucial to have someone else review the manuscript to enhance its readability. The article contains overly lengthy sentences, redundancies, and opportunities for simplification. If you are hesitant to send it to a professional style editor, consider having a colleague review it to make the content more accessible to the reader

Examples of other minor aspects that would help this point:

Use numbers for the separation of the sections, they are easier to follow; for example: 1; 1.1; 1.2; 1.2.1; etc.

Jumps from section I] and to I.A.a), there is no I.A], as in the folowing sections. Anyway, as I commented above, this section should be eliminated or reduced and the numbering should be different.

Ln 81-85: I consider that listing the type of audience is irrelevant, in any case this sentence does not make much sense here, nor does it add much. I would eliminate it, or include it more towards the beginning or the end.

Ln 87-89: should include references

Ln 565: replace “pox virus” by “poxvirus”
Ln 639: remove the square in the sentence
Ln 684: replace “laboratory raised” by “laboratory-raised”

Ln 853: use semicolon to separate the sections (a) (b).... And use the appropriate spaces between them.

Ln 950: Use "total", "global" or "whole" microbiota, to clearly differentiate it from "beneficial microbiota" and prevent sound redundant or repetitive.

Ln 953: Replace “did not mention the developmental stage precisely at which”.... by “did not mention the precise developmental stage at which”

Experimental design

It is a review, but I consider that the bibliography used is up to date and quite abundant.

Validity of the findings

Do not apply

---

## Round 0.2 · Minor Revisions

Thank you very much for having improved the manuscript following all the indications given. However, the reviewers consider that there are still some minor modifications that would improve it. I therefore invite you to make them and resubmit the manuscript to us again.

Reviewer 1 ·

Basic reporting

No comment

Experimental design

No comment

Validity of the findings

No comment

Additional comments

I would like to thank the authors for addressing all my concerns regarding the last version of the paper. I extend my congratulations to the authors for making significant improvements in the content, language, and organization of the paper compared to its previous version. The present version provides a better overall understanding of the fish immune system, microbiota succession, and the ontogeny of the immune system. I recommend accepting the article after making the following minor corrections.

I have only a few minor comments:

Lines 253-256: The sentences are not clear.
Line 300: "Immunology development" seems incorrect.
Line 520: The reference seems out of place and does not follow the required style.
Line 585: The reference appears to be given as a link.
Lines 798-805: These statements need to be substantiated with proper references.

Reviewer 2 ·

Basic reporting

All my comments were addressed.

Experimental design

All my comments were addressed.

Validity of the findings

All my comments were addressed.

Additional comments

All my comments were addressed.

Reviewer 3 ·

Basic reporting

For my part, the document would be ready for publication. I believe it will be a valuable tool for all those interested in the study of fish microbiota and how it affects the immune system of fish.

Just a few comments/suggestions on the text: Ln 347-349; Ln 449, Ln 502 and Fig. 1

Experimental design

NA

Validity of the findings

NA, it is a review

Annotated reviews are not available for download in order to protect the identity of reviewers who chose to remain anonymous.

---

## Round 0.3 · accepted · Accept

Many thanks for improving your manuscript which now can be accepted for publication. Thank you for submitting your work to this journal.

With kind regards,